# Merging short and stranded long reads improves transcript assembly

**Amoldeep S. Kainth**[1]◉, **Gabriela A. Haddad**[2]◉, **Johnathon M. Hall**[1], **Alexander J. Ruthenburg**[1,2,3]*

**1** Department of Molecular Genetics and Cell Biology, The University of Chicago, Chicago, Illinois, United States of America, **2** Committee on Genetics, Genomics and Systems Biology, The University of Chicago, Chicago, Illinois, United States of America, **3** Department of Biochemistry and Molecular Biology, The University of Chicago, Chicago, Illinois, United States of America

◉ These authors contributed equally to this work.
* aruthenburg@uchicago.edu

**Data Availability Statement:** Short- and long-read data generated for this study have been deposited at the Gene Expression Omnibus (GEO) under accession number GSE215355 and GSE215357. A standard script for SLURP and TASSEL is provided

## Abstract

Long-read RNA sequencing has arisen as a counterpart to short-read sequencing, with the potential to capture full-length isoforms, albeit at the cost of lower depth. Yet this potential is not fully realized due to inherent limitations of current long-read assembly methods and underdeveloped approaches to integrate short-read data. Here, we critically compare the existing methods and develop a new integrative approach to characterize a particularly challenging pool of low-abundance long noncoding RNA (lncRNA) transcripts from short- and long-read sequencing in two distinct cell lines. Our analysis reveals severe limitations in each of the sequencing platforms. For short-read assemblies, coverage declines at transcript termini resulting in ambiguous ends, and uneven low coverage results in segmentation of a single transcript into multiple transcripts. Conversely, long-read sequencing libraries lack depth and strand-of-origin information in cDNA-based methods, culminating in erroneous assembly and quantitation of transcripts. We also discover a cDNA synthesis artifact in long-read datasets that markedly impacts the identity and quantitation of assembled transcripts. Towards remediating these problems, we develop a computational pipeline to "strand" long-read cDNA libraries that rectifies inaccurate mapping and assembly of long-read transcripts. Leveraging the strengths of each platform and our computational stranding, we also present and benchmark a hybrid assembly approach that drastically increases the sensitivity and accuracy of full-length transcript assembly on the correct strand and improves detection of biological features of the transcriptome. When applied to a challenging set of under-annotated and cell-type variable lncRNA, our method resolves the segmentation problem of short-read sequencing and the depth problem of long-read sequencing, resulting in the assembly of coherent transcripts with precise 5' and 3' ends. Our workflow can be applied to existing datasets for superior demarcation of transcript ends and refined isoform structure, which can enable better differential gene expression analyses and molecular manipulations of transcripts.

at the GitHub repositories (https://github.com/kainth-amoldeep/SLURP and https://github.com/kainth-amoldeep/TASSEL).

**Funding:** G.A.H. is supported by the NIH Genetics & Regulation Training Grant (T32 GM07197) and the Genetic Mechanisms and Evolution Training Grant (T32 GM139782). This work is supported by NIH grants (R01HL148719 and R35GM145373) to A.J.R. The funders had no role in study design, data collection and analysis, decision to publish, or preparation of the manuscript.

**Competing interests:** The authors have declared that no competing interests exist.

## Author summary

The study of transcriptomes is pertinent to development, disease and response to stimuli. Transcriptomes have largely been studied via short-read RNA-sequencing. Despite its wide use, this technology falls short of illuminating coherent transcripts, particularly for poorly annotated or low-coverage regions. The advent of long-read sequencing has enabled improved analyses of genomes and transcriptomes; however, a systematic direct comparison of short- and long-read RNA-seq and their seamless amalgamation has not been done yet. Here, we demonstrate that short-read RNA-seq provides higher depth for transcript quantitation whereas long-read RNA-seq provides better qualitative information. We report a widespread cDNA synthesis artifact that can markedly impact transcript assembly and quantitation. We develop a computational pipeline to infer strand-of-origin in the long-read cDNA libraries and demonstrate that its application markedly rectifies the erroneous assembly of transcripts. Combining the stranded long reads with the short reads in our new hybrid pipeline, TASSEL, leads to substantial improvement in the assembly of transcripts. This pipeline can be applied to a wide range of datasets, enabling improved characterization for downstream experimentation.

## Introduction

Characterization of transcripts and their variants is essential to advancing our understanding of gene expression in development, responses to stimuli, and disease etiology. The advent of massively parallel short-read sequencing enabled the development of transcriptomics and discovery of biological features such as gene expression levels and pervasive genome-wide low-level transcription [1,2]. High depth of sequencing, relatively low error rates, information about strand of origin and ability to sequence a wide range of RNA from varied upstream sources are some of the key advantages of short-read RNA-seq [3,4]. Furthermore, multiple computational tools have been developed and benchmarked for the alignment, assembly and relative quantitation of short-read transcripts for general as well as application-specific purposes [5,6]. However, short-read sequencing approaches bear several caveats: *i*) library preparation for short-read RNA-seq typically involves PCR amplification which is prone to length and sequence-based biases that can compromise quantification and in some cases detection [7,8]; *ii*) the length of reads (typically 50–200 bp) is much shorter than most of the transcripts, such that transcript assembly and quantitation profoundly relies on inferential re-construction of transcripts; *iii*) differences in the computational algorithms lead to substantial inconsistencies in detection of transcripts and their splicing isoforms, variations in abundance estimation, and potential segmented assembly of a continuous transcript [6,9–12]. These limitations are exacerbated for non-canonical transcripts such as long noncoding RNA (lncRNA) that are typically low abundance and lack canonical features of coding transcripts [9,13].

Towards remediating these issues, new tools have become recently available. Long-read sequencing, also known as third generation sequencing, has permitted longer spans or in some cases full-length transcripts to be sequenced [14]. The MinION device from Oxford Nanopore Technologies (ONT) can directly sequence cDNA which could potentially overcome the segmentation and PCR amplification limitations of short-read sequencing [15]. While the typical read length from short-read sequencing is less than 250 nt, long reads from ONT can reach ≥10kb [16]. This can result in improved and consistent detection of transcripts and their splicing isoforms, better mapping of repeat regions, and potential mitigation of the short-read segmentation problem [17–20]. While long-read sequencing has several advantages, it too has its

share of caveats, being prone to higher error rates [18,21,22], lower transcript coverage and depth [23,24], lack of strand of origin information for reads obtained from methods such as ONT direct and PCR cDNA sequencing, and having a more limited toolkit for analysis in comparison to short-read sequencing. As many of the analysis tools were developed for DNA-level genomic assembly, they are ill-equipped to detect variations in transcript copy number and isoforms [25]. Although an array of tools has been developed for long-read RNA-seq analysis [catalogued in 26], many of them are still in their infancy and lack robustness [27]. Hybrid algorithms developed to combine short- and long-read datasets are largely built either for error correction in the long-read data [21,28] or employ short-read as a template for long-read assembly [29]. Consequently, the output is reduced in long-read information with bias towards short-read attributes [28]. There remains a dearth of approaches that truly integrate short- and long-read datasets for transcript calling. Critically, assembly frameworks of many of the current approaches are modeled on well-annotated high abundance canonical transcripts. Further refinement and development of long-read transcriptomic tools is warranted for analyses of non-canonical transcripts such as lncRNA which constitute a large fraction of the transcriptome [30].

LncRNA have emerged as important regulators of mammalian genomes, implicated in the regulation of genes through a diverse and growing set of mechanisms [31–36]. One class of transcripts that are highly enriched in lncRNA are the recently described chromatin-enriched RNA (cheRNA) [37–40]. CheRNA have been particularly difficult to characterize as they are highly cell-type divergent, predominantly unannotated, and so low abundance that they were largely unobserved prior to biochemical enrichment [37,38]. Owing to these features, we reasoned that cheRNA would provide a challenging testbed to develop and benchmark long-read RNA-seq analyses for assembly and quantitation of transcripts.

In this study, we improve existing bioinformatic workflows to integrate the quantitative depth of short-read sequencing with the qualitative strengths of long-read sequencing for improved characterization of transcripts. Using short- and long-read datasets we generated in two cell lines with mono-exonic ERCC spike-in standards, as well as existing datasets of multi-exonic sequin spike-in standards, we systematically compare key parameters and biases in the read alignment and assembly of transcripts. We also report novel artifacts and further document known pitfalls in long-read datasets that can substantially impact the identity and abundance of assembled transcripts. We develop a computational pipeline, SLURP (Stranding Long Unstranded Reads using Primer sequences), to infer strand-of-origin from ONT long-read cDNA libraries (hereafter "stranding"), markedly improving assembly of transcripts from long reads. Incorporating our stranding method, we devise a hybrid transcript assembly pipeline, TASSEL (Transcript Assembly using Short and Strand Emended Long reads), that merges qualitative features of stranded long reads with the quantitative depth of short-read sequencing. TASSEL outperforms other assembly methods in terms of sensitivity and complete assembly on the correct strand. At the molecular level, TASSEL resulted in substantially improved capture of key transcriptomic features such as transcription start and termination sites as well as better enrichment of active histone marks and RNA Pol II. To demonstrate the efficacy of TASSEL, we implement it for improved characterization of cheRNA, resolving previously ambiguous transcripts into coherent and discrete molecules. SLURP is generally applicable to other cDNA-based ONT long-read datasets and TASSEL can be applied to ONT as well as PacBio long-read RNA-seq to refine transcript identification and isoform structure, which can enable better molecular manipulations to illuminate their mechanisms.

## Results

### Using short- and long-read RNA-seq to capture chromatin-enriched RNA

To evaluate the merits of short- and long-read sequencing platforms for assembly and molecular feature characterization of transcripts, we focused our efforts on a specific but challenging class of transcripts called cheRNA (chromatin-enriched RNA). We leveraged biochemical fractionation of nuclei [37,41] to obtain cheRNA (Fig 1A) in two cell lines—HAP1, a near-haploid leukemia cell line derived from KBM-7 [42], and HL1, a cardiac muscle cell line derived from AT-1 mouse atrial cardiomyocytes [43]. Importantly, at the nuclear extraction step, mono-exonic ERCC RNA standards [44] were spiked in to allow for quality control analyses and direct comparisons between samples for both short- and long-read libraries. Fractionated RNA was then depleted of ribosomal RNA and subjected to either short-read or long-read RNA sequencing. Owing to higher quantitative depth, short-read sequencing (n = 3) was done on both fractions to identify transcripts which are significantly enriched in the chromatin fraction as compared to the nucleoplasmic fraction. We define these transcripts as cheRNA (S1A Fig). We performed long-read sequencing (n = 2) for only the chromatin fraction for comparison and conjugation with short-read transcripts from the chromatin fraction and to refine cheRNA transcript models. For long-read RNA-seq, RNA was polyadenylated for use with polyT primers, subjected to library preparation using the Oxford Nanopore direct cDNA sequencing kit, then run on a MinION sequencer. We chose the cDNA sequencing platform as it has higher yield than the direct RNA sequencing platform [45]. We obtained ~1.2M and 3M reads for HAP1 replicates, and ~1.8M and 2.5M reads for HL1 replicates and subjected them to our analysis pipeline (S1 Table). The median sizes of reads in the long-read libraries were 1.7–1.9kb for HAP1 samples and 1.2–1.4kb for HL1 samples; replicates were nearly identical in read length profiles (Figs 1B and S1B and S1 Table). Thus, our long-read sequencing captures much longer reads than typical short-read sequencing. Next, long reads were aligned using minimap2 [46] to human (hg38) and mouse (mm10) genome assemblies for HAP1 and HL1, respectively. Aligned long reads showed strong correlation for genomic coverage between the replicates (pearson r = 0.92 and 0.86 for HAP1 and HL1, respectively) (Figs 1C and S1C). Similarly, short-read alignments with hisat2 [47] also showed high correlation between replicates (S1D Fig) for both cell lines, attesting to the reproducibility of the RNA fractionation, library preparation, and alignment methodology used here.

### Spike-in standards highlight similarities and limitations in short- and long-read alignment

Transcript coverage is critical for correct assembly and estimation of abundance. The use of ERCC spike-in RNA enables us to directly compare coverage of short- and long-read alignments. Short-read alignments were highly variable in the gene body with clear lack of coverage at the 5' and 3' ends, whereas long-read alignments showed largely homogenous coverage across ERCC transcripts (Figs 1D and S1E). Next, we tested the extent to which a given transcript is covered as a function of its abundance or length. We observed that the fraction of a given transcript covered by short-read alignment showed a sigmoidal relationship with the amount added: near-linear sensitivity for low abundance transcripts, with coverage saturating at mid- to high-abundance transcripts (Figs 1E and S1F). While long-read alignments also showed a similar trend, they had less dynamic range, especially for low abundance transcripts. Of the 92 ERCC transcripts, 78 and 82 were detected for HAP1 and HL1 short-read datasets respectively (Figs 1E and S1F, *left*), as compared to 51 and 58 in the long-read datasets (Figs 1E and S1F, *right*). This is consistent with higher depth of the short-read datasets as compared to

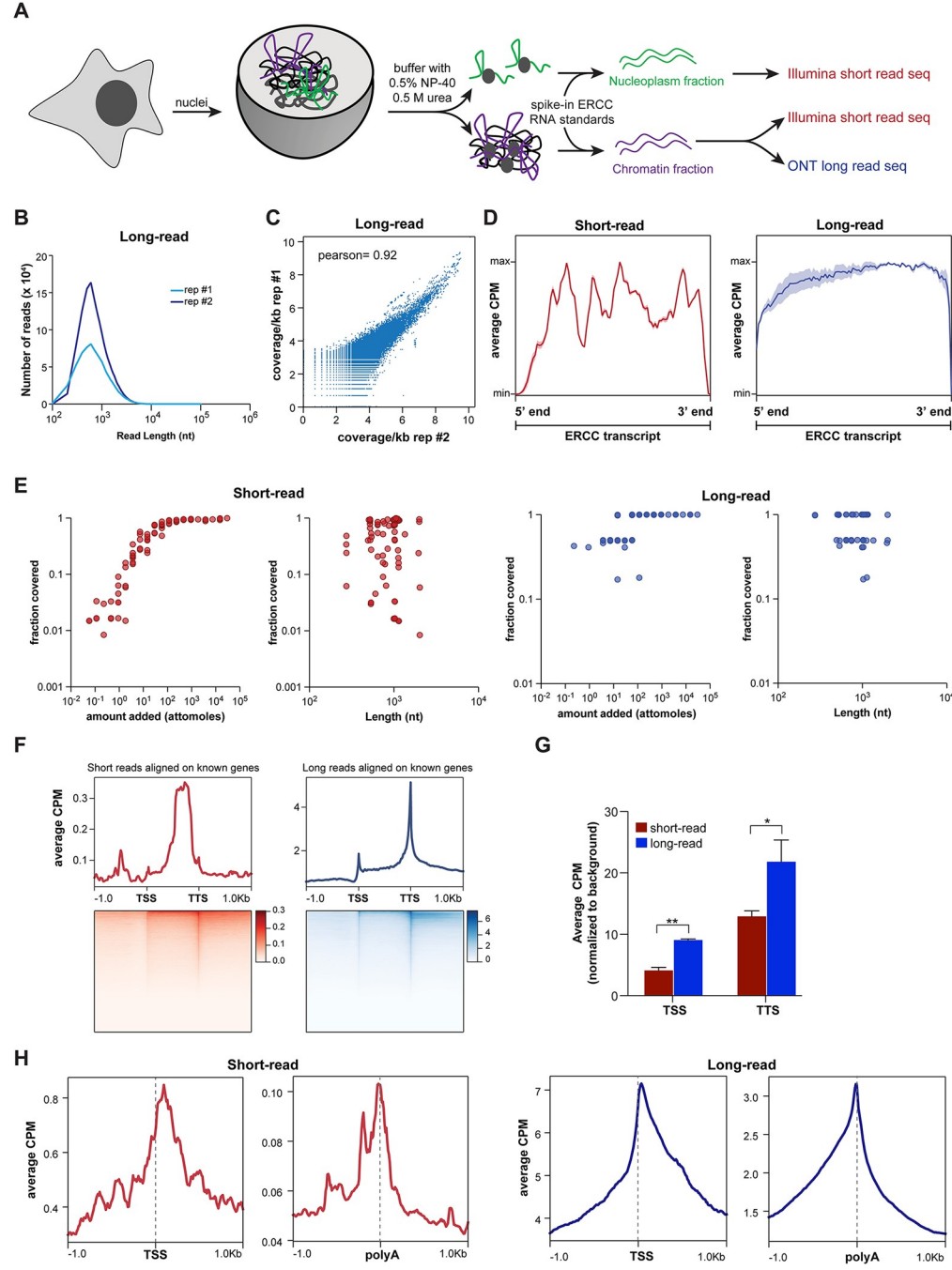

**Fig 1. Short- and long-read RNA-seq to capture chromatin-enriched RNA. A.** Schematic illustrating the nuclear fractionation of cells to isolate chromatin-enriched RNA (cheRNA) then subjected to short- and long-read sequencing. Nuclei from HAP1 and HL1 cells were fractionated into nucleoplasm and chromatin fractions using urea/detergent buffer [37], and ERCC standard RNA [44] were spiked in before RNA extraction and sequencing. For short-read, both the nucleoplasm and chromatin fraction RNA were subjected to single end 50 bp Illumina sequencing (n = 3). For long-read, only the chromatin fraction was sequenced using Oxford Nanopore Technology (ONT; n = 2). **B.** Density plot of read lengths from long-read sequencing of HAP1 replicates (n = 2). **C.** Correlation between HAP1 replicates (n = 2) for genomic coverage of long-read alignments binned at 1 kb. Pearson correlation is shown. **D.** Average counts per million (average CPM, dark color; +/- SD, lighter color) of aligned reads across all ERCC transcripts, meta-scaled to 1000 nt, in the HAP1 short-read (n = 3) and long-read (n = 2) samples. **E.** Relationship between the average fraction of ERCC transcripts covered by short (left) or long (right) read alignments as a function of their amount (attomoles) or length (nt) for HAP1 samples. **F.** Metagene plots and heatmaps of mapped read coverage (average CPM) for HAP1 short- and long-read alignments, scaled to transcription start sites (TSS) and transcription termination sites (TTS) of

gencode hg38v41 genes. **G.** Enrichment of TSS ± 50 bp as well as TTS ± 50 bp in short-read replicates (n = 3) and long-read replicates (n = 2) in HAP1 samples. Average counts in the indicated region for each of the replicates were normalized to a background of random 100 bp regions in the same library to account for variations in the depth of the libraries. * p < 0.05; ** p < 0.01; unpaired Student's t-test. **H.** Metagene plots of mapped read coverage (average CPM) for HAP1 short (left) and long (right) reads, centered on curated transcription start sites (TSS; [50]) or termination sites (polyA; [51]).

the long-read datasets (S1 Table). Importantly, the fraction coverage of ERCC transcripts was independent of the length of the transcripts for both short- and long-read alignments. Similarly, the coverage of known genes in our HAP1 and HL1 datasets is also independent of the length of the gene (S1G Fig), indicating that our libraries are free of any artifactual length biases. Thus, high correlation between replicates, sensitivity to abundance, and absence of length bias demonstrate the reliability of our library-prep and alignment methods for both sequencing platforms.

### Long-read alignments are better in capturing transcript termini

As short-read libraries are generated using random hexamers from fragmented RNA-samples and sequenced often with read lengths of less than 100 bp, a given read covers only a fraction of the transcript. Since long reads have the potential to capture a full-length transcript (Figs 1B and S1B), we reasoned that long-read alignment might be able to capture ends of transcripts better than short-read alignment. To formally test this hypothesis, we compared the distributions of long- and short-read alignments over known genes. While the short-read alignments showed better coverage in the gene bodies, the long-read alignments evinced better demarcation of transcription start sites and transcription termination sites of annotated genes (Figs 1F, S1H and S1I). A direct comparison showed that the 5' and 3' ends of a gene are more prevalent in the long-read alignments than the short-read alignments (Fig 1G). The lower gene-body coverage in the long-read alignments is consistent with previous findings [20,48]. Additionally, we observed strong 3'-end coverage bias for long reads as reported by previous studies [20,49].

A limitation of the foregoing analyses is that they are restricted to reference genes. However, there are many more experimentally derived TSS and TTS independent of the consensus reference annotation. To expand our analysis, we calculated the coverage of short- and long-read alignments on transcription start sites annotated on the basis of CAGE, TSS-Seq and RAMPAGE [50] as well as highly curated polyA sites [51]. Alignments from both sequencing platforms showed enrichment over these key ends of transcripts (Figs 1H, S1J and S1K). The metagene plots of long-read alignments centered over curated TSS showed more density going into the gene body relative to upstream of the TSS for both cell lines. Conversely, long-reads centered over TTS showed more density coming from the gene body relative to downstream of the TTS for both cell lines. Corresponding short-read plots showed noisier profiles and symmetrical density distributions, indicating less biologically representative capturing of 5' and 3' ends (Figs 1H, S1J and S1K). Thus, we conclude that long reads are capturing transcript molecule ends more effectively than short reads. Enhanced capture of intact transcripts in long-read sequencing can improve the detection of rare isoforms and gene-fusion events (see S1 Note). Taken together, these analyses suggest that short- and long-read datasets can complement each other for better coverage of transcriptomic elements.

### Benchmarking long-read transcript assembly

One of the crucial steps of RNA-seq analysis is assembly of transcripts from the reads as it dictates the identification of transcript structure, isoform variants, gene abundance and

differential gene expression. Therefore, we compared three different transcript assembly pro-
grams: Cufflinks [52], StringTie2 [53,54] and FLAIR [55]. First, we compared these programs
for the assembly of the ERCC spike-in standards as we know the exact transcript structure of
these transcripts. We observed that guided assembly with StringTie yielded highest sensitivity
and precision across short- and long-read assembly (S2A Fig). A major limitation of the ERCC
spike-in standards is that they are unspliced transcripts with no alternative isoforms; as such
they may fail to capture the complexity of a typical mammalian transcriptome. To overcome
this caveat, we also tested the assembly methods on the sequin spike-in standards in the exist-
ing short- and long-read datasets from the SG-Nex consortium [48]. Sequins are multi-exonic
spike-in RNA standards with alternative isoforms [56]. Similar to ERCC standards, guided
assembly with StringTie outperformed other short- or long-read methods (S2A Fig). Further-
more, StringTie also performed better than other methods for transcript assembly in our
HAP1 and HL1 datasets. In general, we observed that short-read transcript assembly has
higher accuracy as well as precision than long-read transcript assembly. This could arise due to
multiple factors such as lower depth, higher error rate and suboptimal assembly of long-read
sequencing [57].

As estimation of transcript and gene abundance is one of the desired outcomes of most
RNA-seq analyses, we assessed whether our short- and long-read alignment and transcript
assembly are sensitive to variations in abundance and lengths of transcripts in the transcrip-
tome. To do so, we computed abundance of mono-exonic ERCC and multi-exonic sequin
spike-in standards, as their identity, abundance, and lengths are known *a priori*. We observed
that the abundance of each spike-in standard in the short- and long-read libraries via StringTie
assembly and quantitation showed a near-linear relationship with the concentration of each
standard transcript added during library preparation (Figs 2A and S2B). A similar relationship
was also observed using BedTools [58] which computes coverage based on alignments and is
independent of transcript assembly (S2C Fig). Short-read library preparation involves PCR
amplification whereas the long-read direct cDNA libraries were made without PCR. The linear
range of transcript quantitation by short-read assembly and its similarity with the long-read
assembly suggests that it is largely free of PCR biases typically thought to be associated with
short-read RNA-seq analyses [7,8]. We also observed that quantitation of short- and long-read
data are independent of the length of spike-in transcripts (Figs 2A and S2B). This is particu-
larly informative for short-read RNA-seq as it demonstrates that although individual RNA is
fragmented during short-read library prep, StringTie can efficiently assemble and quantitate a
wide range (0.2–7 kb) of well-annotated transcripts from short reads. However, in the absence
of an adequate ground truth knowledge of the precise transcriptome in these cell lines, we cau-
tion the extrapolation of this finding to more complex and diverse mammalian transcriptomes.
Assembled transcript counts from long-read replicates were well correlated for medium and
high abundance transcripts (Fig 2B, HAP1 $R^2$ = 0.83; S2D Fig, HL1 $R^2$ = 0.74), indicating
reproducibility of the extraction, sequencing, and analysis pipeline. Consistent with higher
counting error at the lower ends of the distribution, we detected increased divergence in quan-
titation of low abundance transcripts.

Having established the efficacy of StringTie in the assembly of transcripts from long reads,
we compared the assembled long-read transcriptomes with reference transcriptomes (Fig 2C).
We observed that more than 80% of long-read transcripts overlapped with the reference tran-
scriptome: 5–14% of transcripts matched exactly with the reference transcriptome, while 6–8%
showed a match of at least one intron-exon junction. In contrast, 45–60% of transcripts assem-
bled from short-read samples showed an exact match with the reference transcriptome (S2E
Fig). Similar to previous analyses [54], we observed that a large percentage of the long-read
assembled transcripts (>40%) were fully contained in reference gene introns. These truncated

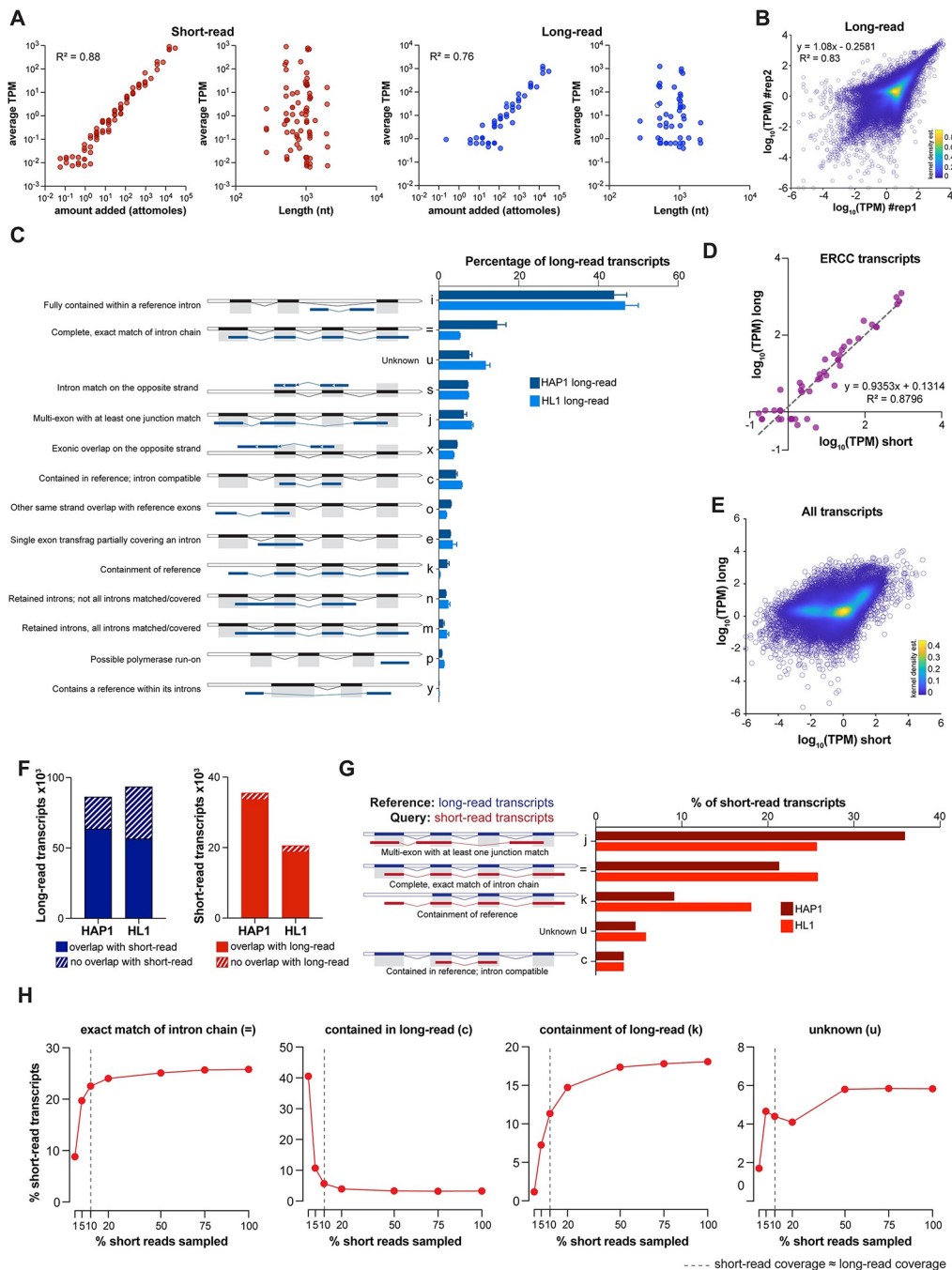

**Fig 2. Optimization of short- and long-read transcript assembly. A.** Average abundance (Tags Per Million; TPM) of ERCC transcripts determined by StringTie versus amount added (attomoles) or length (nt) of the transcript in HAP1 samples (n = 3 for short reads, n = 2 for long reads). Short-read plots are shown in red (left), long-read plots in blue (right). $R^2$, correlation coefficient for linear regression. **B.** Correlation between HAP1 replicates for long-read transcriptome assembly by StringTie. Shown are the scatter plots of abundance (TPM) of each transcript in the two replicates. Colors correspond to kernel density estimations of scatter plot distribution. $R^2$, correlation coefficient for linear regression. **C.** Comparison of structure of transcripts assembled by StringTie for HAP1 and HL1 long-read samples with structure of reference genome transcripts. The class codes for relationship between the assembled transcript and the closest reference transcript were deduced from gffcompare. **D.** Correlation plot for the average abundance (TPM) of ERCC spike-in transcripts in the short-read datasets (n = 3) with the average abundance in the long-read datasets (n = 2) of HAP1 samples, determined by StringTie. Shown are the transcripts detected by both sequencing platforms. **E.** Scatter plot comparing the average abundance (TPM) of all transcripts in the short-read datasets (n = 3) and the long-read datasets (n = 2) in HAP1 samples. Transcripts assembled by short- and long-read

assembly were merged using StringTie merge to obtain a non-redundant pool of transcripts across samples. Abundance in each sample was then calculated using the re-estimation function of StringTie using this merged transcriptome as the template. Color bar corresponds to kernel density estimations of scatter plot distribution. **F.** Number of long-read transcripts with or without any overlapping transcript in the short-read datasets (*left*); number of short-read transcripts with or without any overlapping transcript in the long-read datasets (*right*). **G.** Structural comparison of short-read assembled transcripts with the long-read assembled transcripts for HAP1 and HL1 samples. The long-read transcriptome assembly was used as the reference and short-read as the query in gffcompare. **H.** Percent of transcripts in the down-sampled short-read dataset for several key transcript structures (class codes in panel 2G) using long-read transcripts as a reference in HL1 samples. Note that the depth of 10% of HL1 short-read dataset (dotted line) is approximately equivalent to the depth of long-read dataset in terms of number of mapped bases.

transcripts could originate from inherent limitations of long-read sequencing, such as incomplete assembly due to low coverage and internal priming of oligo-dT primers [reviewed in 20]. The prevalence of such transcripts is higher in our dataset, possibly because of the enrichment of nascent transcripts in the chromatin-bound transcripts [59–61]. Notably, many transcripts in our long-read datasets (~7%; Fig 2C class code "s"), in contrast to the short-read datasets (S2E Fig, class code "s"), were assembled to the opposite strand of the reference transcriptome. This is because strand information is lost during the preparation of cDNA for ONT direct cDNA long-read RNA-seq. In contrast, the short-read libraries were prepared using a method that preserves the strand information.

## Comparison of short- and long-read data

After establishing that our long-read datasets pass initial quality control measures, we assessed long-read sequencing performance relative to the short-read sequencing. To do so, we leveraged matching short-read datasets for each cell line, which also contained spiked in ERCC RNA. Analysis of averaged TPMs of ERCC standards from StringTie for long reads (n = 2) against short reads (n = 3) showed strong correlation between the HAP1 and HL1 datasets ($R^2$ = 0.88 and 0.91, respectively) (Figs 2D and S2F). Expanding this analysis to all transcripts assembled by StringTie showed high correlation of quantitation for highly abundant transcripts (Figs 2E and S2G). However, we observed increased divergence for low abundance transcripts ($\log_{10}(\text{TPM}) < 0$), as short-read sequencing showed more dynamic range than long-read sequencing for these transcripts. This is consistent with higher sensitivity of short-read sequencing for low abundance transcripts (Figs 2A and S2B). While both transcript assembly platforms peaked at similar lengths in their distributions, short-read assemblies had a higher number of long transcripts (S2H Fig).

We observed more than 70% of long-read transcripts in the HAP1 dataset and more than 60% in the HL1 dataset had an overlapping transcript in the corresponding short-read dataset (Fig 2F). Conversely, more than 90% of short-read transcripts had an overlap in the corresponding long-read dataset (Fig 2F). Then, we compared the fine-scale structure of short- and long-read assembled transcripts. More than 20% of short-read transcripts in the HAP1 dataset and more than 25% of short-read transcripts in the HL1 dataset showed an exact match with their long-read counterparts (Fig 2G). Additionally, ~30% of multi-exon short-read transcripts had at least one intron-exon junction matched with the corresponding long-read transcripts. The differences in transcript structure between the two platforms could arise due to differences in the depth of sequencing or the inherent differences in the sequencing and assembly methodologies. To test the effect of depth differences, we compared the long-read transcriptome structure with that derived from down-sampled short reads (Fig 2H). The percentage of short-read transcripts matching exactly with a long-read transcript increases concomitantly with the depth of short-read sequencing (Fig 2H, class code "=") until the short-read coverage equals that of long-read coverage. We detected minimal change thereafter as it denotes the maximum

limit of coherent assembly for long-read assemblies and saturation of the total number of transcripts assembled in the short-read assembly (S2I Fig). The inverse trend is seen for short-read transcripts that are contained within the long-read transcripts (Fig 2H, class code "c"), emphasizing the fact that assembly from low-depth short-read RNA-seq leads to artifactual segmentation of transcripts. Conversely, we detected that a higher proportion of short-read transcripts contained a long-read transcript with increase in the depth of short-read sequencing (Fig 2H, class code "k"). This suggests that lower depth of long-read sequencing can cause incomplete assembly of transcripts, consistent with our observation that short-read assemblies had a higher number of long transcripts (S2H). The proportion of short-read transcripts with no counterpart in the long-read dataset was typically higher in the high-depth short-read dataset (Fig 2H, class code "u"). These analyses indicate that sequencing depth is an important but not the only factor that limits concordant assembly of transcripts between the short- and long-read platforms.

Our analyses highlight two main caveats of transcript assembly by long reads: assembly of long-read transcripts on the incorrect strand due to the lack of strand of origin information, and containment of long-read transcripts in the reference genes due to the lack of sequencing depth. We undertook to develop a computational pipeline to overcome these limitations.

## Development of computational stranding of long reads to improve transcript assembly

There are many methods to prepare and sequence stranded libraries for short-read RNA-seq [62] which culminate in accurate assembly and analyses of transcripts [63]. However, libraries prepared from ONT direct and PCR-cDNA methods inherently lack strand of origin information. Consequently, these methods suffer from alignment of reads to the incorrect strand of the locus of origin, leading to assembly of transcripts opposite to their true strand (Fig 2C). As the existing software for stranding of long reads discarded >70% of the reads in our datasets (S3A Fig), we sought to develop a computational "stranding" pipeline, SLURP (Stranding Long Unstranded Reads using Primer sequences). Our method exploits the sequences of primers used during preparation of cDNA long-read libraries to infer strand of origin (Fig 3A). Specifically, primer 1 (with a polyT 3' end) is used to synthesize the first strand of cDNA by reverse transcriptase, which subsequently switches its template strand using primer 2 (with a GGG 3' end) to synthesize the second cDNA strand. Therefore, we reasoned that the presence of primer 1 or the reverse complement of primer 2 in a read would indicate that it originates from the first strand, whereas the presence of primer 2 or the reverse complement of primer 1 would denote that the read is derived from the second strand. As the primers are incorporated in the majority of the reads, we took advantage of a motif-enrichment tool, MEME [64], to predict the sequences of primer 1 and primer 2 (S3B Fig). We checked the location and prevalence of the two primer sequences in the long reads. Consistent with priming from the transcript termini, primer 1 and primer 2 were highly enriched within the first 100 bp of the reads while their reverse complements were enriched in the last 100 bp of the reads (Fig 3B). To our surprise, we observed that the prevalence of primer 1 (>53% of reads) is considerably higher than primer 2 (<9%) in the direct-cDNA kit, suggesting that the first strand synthesis step is more efficient than strand-switching and second strand synthesis. This skew may account for the coverage bias at the 3'-end of long-read RNA-seq reported in many previous studies [reviewed in 20].

We compared different combinations of primer searches as well as permissible number of mismatches to strand our long-read libraries. Guided by their prevalence (Fig 3B), we restricted our search of primer sequences to the first or last 100 bp of reads to minimize

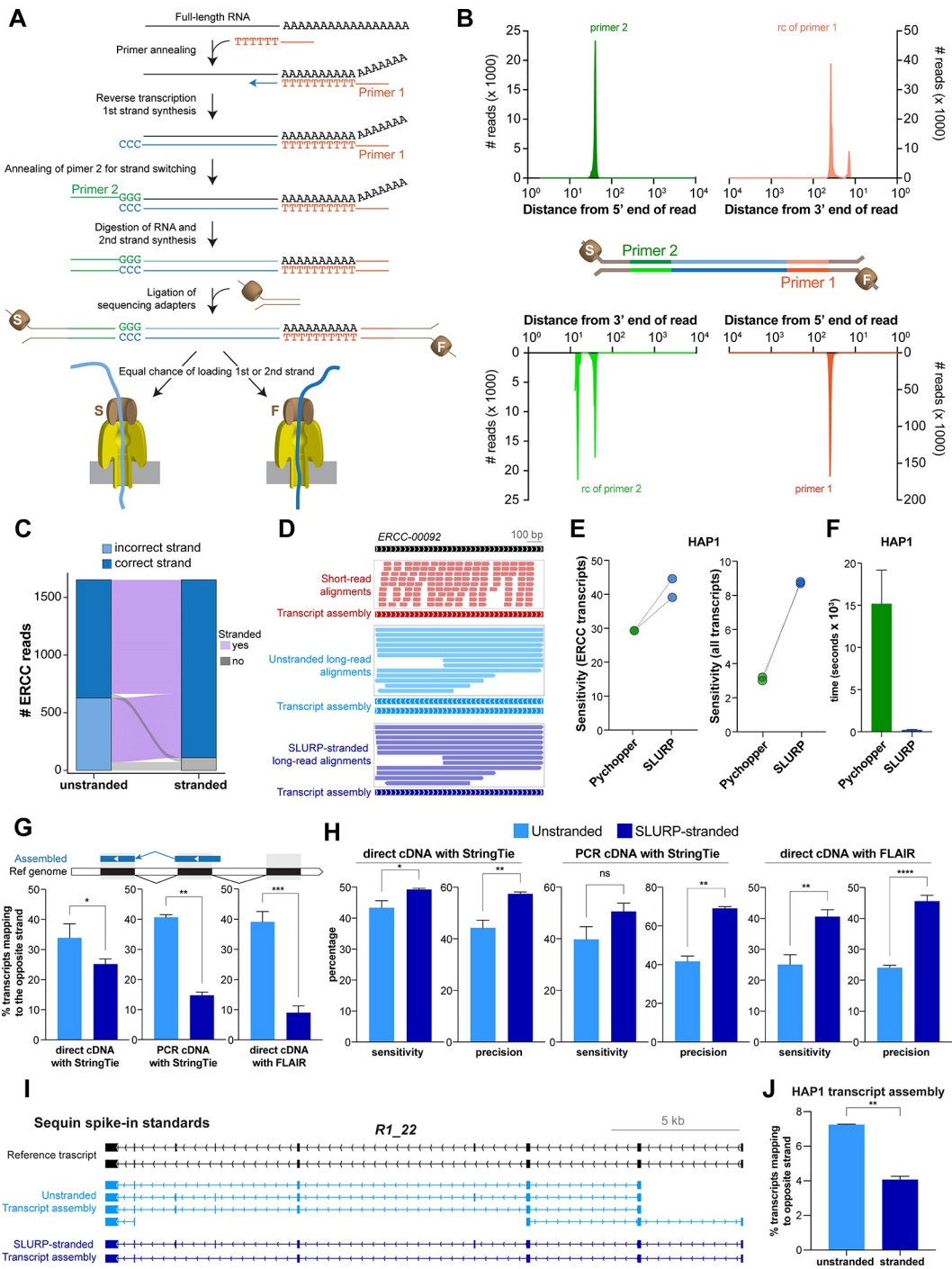

**Fig 3. Primer-based elucidation of the strand of origin of long reads. A.** Schematic of ONT long-read cDNA library preparation. Primer 1 with 3' polyT initiates first strand synthesis by reverse transcriptase that adds non-templated -CCC- at the end. Primer 2 with -GGG- end anneals to the first strand and mediates strand switching for synthesis of the second strand. Sequencing adapters are subsequently ligated to the double stranded cDNA molecule. Either of the two strands (F: first, S: second) can be loaded in a given sequencing nanopore. **B.** Location and enrichment of sequencing primers (primer 1 and primer 2) and their reverse complements in the long reads. **C.** Change in the correct strand mapping of unstranded and stranded long-reads to ERCC transcripts. **D.** Assembly of ERCC-00092 transcript by unstranded and stranded long reads. **E.** Comparison of Pychopper and SLURP for the sensitivity of assembling the ERCC (left) or human (right) transcripts in the HAP1 dataset. **F.** Comparison of user time required by Pychopper and SLURP to strand reads in the HAP1 dataset. **G.** Comparison of percentage of sequin spike-in standard transcripts mapping to the opposite strand of the reference annotation in the unstranded and stranded long-read assemblies using direct cDNA and PCR-cDNA datasets with StringTie as well as

direct cDNA dataset with FLAIR (n = 2–3; mean ±SD; * p < 0.05; ** p < 0.01; *** p < 0.001 unpaired Student's t-test). **H.** Comparison of SLURP-mediated change in the sensitivity and precision of the sequin spike-in standards transcript assembly. **I.** Example of a sequin spike-in standard transcript correctly assembled upon SLURP-mediated stranding of long reads. **J.** Comparison of percentage of transcripts mapping to the opposite strand of reference transcripts in the unstranded and stranded long-read transcriptome of HAP1 cells (n = 2; mean ±SD; ** p < 0.01 unpaired Student's t-test).

artifactual stranding due to coincidental occurrence of these sequences in read interiors. We observed that a 3-criteria (primer 1, primer 2 and reverse complement of primer 2) search combined with 2 mismatches provided an appropriate balance between the number of assembled transcripts and extent of correct stranding with respect to the reference transcriptome (S3C and S3D Fig). SLURP yielded significantly more stranded reads than UNAGI [65] and Pychopper (https://github.com/epi2me-labs/pychopper) (S3A Fig). We checked the efficacy of SLURP on ERCC spike-in standards, as their precise strand of origin is known *a priori*. SLURP successfully reassigned the majority (>90%) of the long reads originated from ERCC loci to their correct strand (Fig 3C). Importantly, while the alignment of unstranded long reads led to ambiguous assembly of two ERCC-00092 transcripts in opposite directions, alignment of stranded long reads led to assembly of one correct transcript (Fig 3D). Notably, SLURP demonstrated higher sensitivity of ERCC transcript assembly than Pychopper, with equivalent or better precision (Figs 3E and S3E) in the HAP1 dataset. Similar increase in sensitivity of detection was also observed for human transcripts in the HAP1 datasets (Figs 3E and S3E). As a result, use of SLURP led to assembly of a substantially greater number of matching transcripts than Pychopper (S3F Fig). We reason that the increase in the yield of stranded reads by SLURP results in increased sensitivity of detection as well as precise end-to-end assembly. For example, use of Pychopper led to a truncated assembly of *ERCC-00108* transcript whereas use of SLURP led to assembly of the full transcript (S3G Fig). Furthermore, use of SLURP had markedly lower computational load than Pychopper (Fig 3F). SLURP-mediated improvement is not limited to StringTie as similar increase in sensitivity and precision was also detected for transcripts assembled by FLAIR (S3H Fig). When tested on HL1 (mouse transcripts) and HCT116 datasets from the SG-Nex consortium, SLURP outperformed Pychopper in terms of sensitivity and precision of assembly as well as the computational overhead of stranding (S3I and S3J Fig).

Then we tested SLURP on multi-exonic sequin spike-in standards. Use of SLURP markedly reduced the proportion of transcripts assembled on the wrong strand of the reference annotation for both direct cDNA and PCR-cDNA kits of the ONT platform (Fig 3G). In addition to the transcript assembly by StringTie, SLURP also rectified wrong-strand assembly by FLAIR. SLURP not only corrected the strand of transcript assembly but also improved the overall sensitivity and precision of the sequin transcriptome in the direct- and PCR-cDNA datasets with StringTie and FLAIR (Fig 3H). For example, neither of the two isoforms of the multi-exonic sequin transcript *R1_22* were correctly assembled using the uncorrected long reads (Fig 3I). SLURP correction led to assembly of both the isoforms in the correct direction. When tested on sequin spike-in standards with alternate TSS/TTS, SLURP improved the sensitivity and precision of their assembly (S3K and S3L Fig). We expanded our analyses to all assembled transcripts in our HAP1 dataset as well as a K562 cell line dataset from the SG-Nex consortium [48]. Within the limits of imperfect annotation of the human reference genome for any given cell line, we observed that the stranding of long reads also led to significant reduction of transcripts assembled to the opposite strand of the reference transcriptome (Figs 3J and S3M). Removal of unstranded reads reduces the erroneous assembly of transcripts (S3N Fig). As exemplified by a known transcript (S3O Fig), the overall reduction in erroneous mapping can lead to assembly of a multi-exonic transcript that matches the gencode isoform. The foregoing

analyses demonstrate that our stranding pipeline can be used as a general tool to infer strand information of complex transcripts from long-read cDNA data independent of short-read datasets or assembly programs to increase the accuracy of transcript assembly and any subsequent analysis.

## Long-read libraries are prone to reverse transcriptase cDNA synthesis artifacts

As per the library preparation method, primer 1 initiates the synthesis of the first strand and primer 2 initiates the synthesis of the second strand after the reverse transcriptase switches strands (Fig 3A). Thus, the occurrence of primer 1 and its reverse complement should be mutually exclusive. Surprisingly, we discovered that many long reads have the primer 1 sequence in their first 100 bp and the reverse complement of primer 1 in their last 100 bp (Fig 4A). The occurrence of the primer 1 sequence at each end of a single read would not be anticipated from the way the library was prepared, yet this is observed in ~6% of total reads (Fig 4B). To test the prevalence of this artifact, we probed a published long-read data set [48] and detected similar levels of such reads (Fig 4B). Although Pychopper indicates the presence of such artifactual occurrences of primer configuration, the possible basis, the underlying sequence features and the downstream effects of such reads on read alignment and transcript quantitation is not known. Further examination revealed that these reads map to the same genomic locus twice in opposite directions (Fig 4C). Notably, we found that one of these two alignments is flagged as a "supplementary alignment" by the minimap2 aligner while the other is considered as the primary alignment. We reasoned that such an alignment could be possible if the reads are palindromic. Indeed, secondary-structure prediction shows that most of these reads are largely palindromic within base calling error, with one half of the read being the reverse complement of the other (Figs 4D, 4E and S4A). Surprisingly, filtering of supplementary alignments led to substantial changes in the computation of coverage and abundance of transcripts (Fig 4F). We speculate that these reads originate when the reverse transcriptase switches strands due to micro-homology [66] and continues to synthesize cDNA using the first strand as the template (Fig 4G), leading to artifactual doubling of read length via reverse complement synthesis. Consistent with this notion, we detected virtually no palindromic reads in the direct RNA long-read libraries that do not use reverse transcriptase for cDNA synthesis (Fig 4B). Hence, these analyses reveal a possible cause and consequences of a widespread artifact in ONT long-read cDNA libraries that contribute to error in read length and counting. The mapping error by minimap2 is propagated to transcript assembly and quantitation by StringTie, which could negatively impact the accuracy of gene expression analyses.

## Integration of short-read data with stranded long-read data improves transcript assembly

To overcome the limitations of long-read transcript assembly, we developed a hybrid transcript assembly pipeline, TASSEL (Transcript Assembly using Short and Strand Emended Long reads), that incorporates long range information of stranded long reads with high depth of short-read sequencing, with very low additional computational burden (Fig 5A). First, we compared TASSEL to other hybrid and standalone long-read transcript assembly programs for transcript assembly of ERCC spike-in standards (Figs 5B and S5B). TASSEL showed markedly higher sensitivity and precision of ERCC transcript assembly when compared to FLAIR [55], Bambu [67], StringTie Mix [29], RNA-Bloom2 [68], IsONform [69], IsoQuant [70] and RATTLE [71]. While the other pipelines failed to assemble >40 of the 92 ERCC transcripts, only 13 transcripts were not assembled via TASSEL (S5B Fig). FLAIR and StringTie

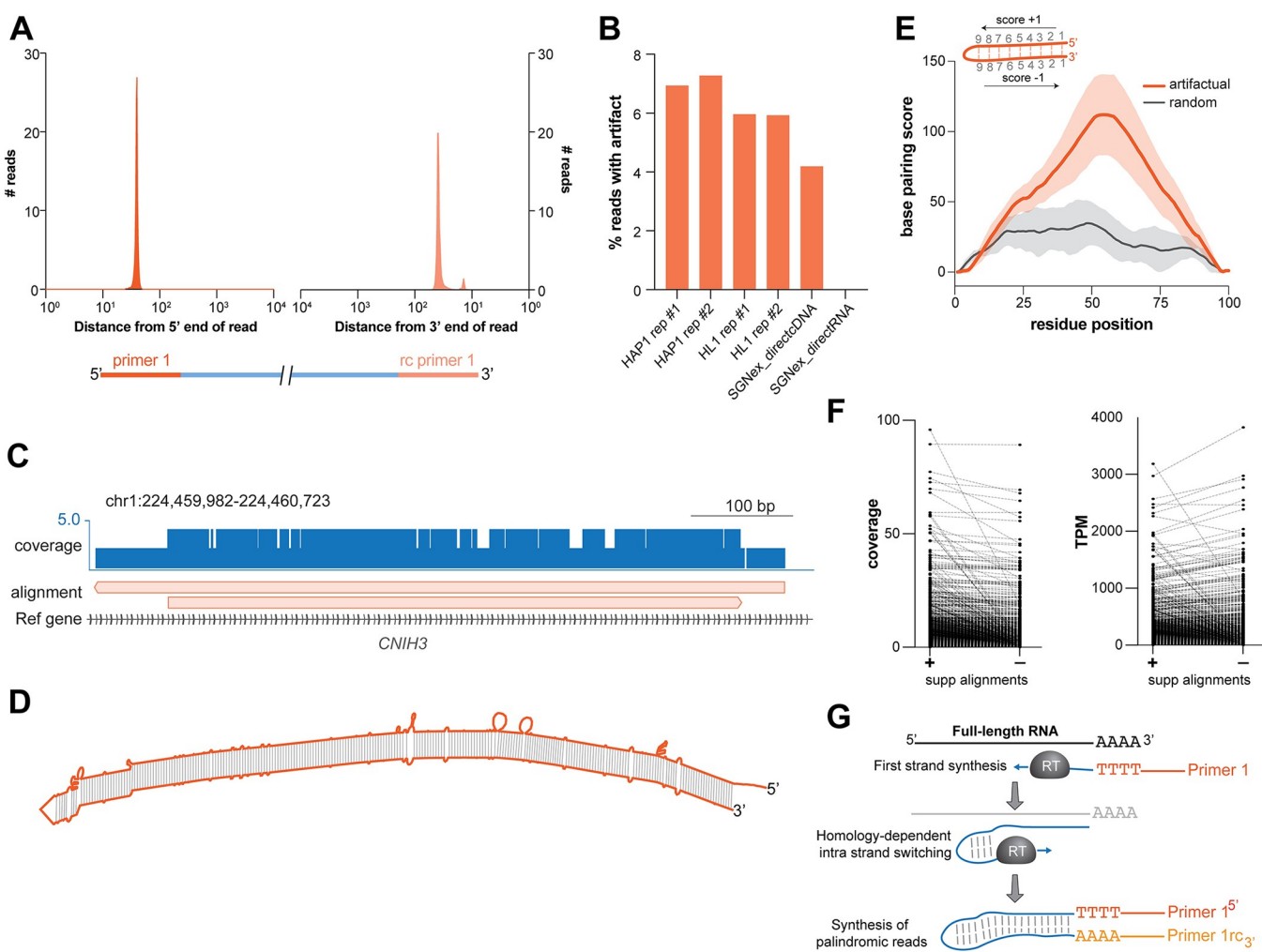

**Fig 4. Widespread cDNA synthesis artifact in long reads. A.** Plot showing enrichment of primer 1 at the 5' end and the reverse complement of primer 1 at the 3' end of artifactual reads. **B.** Prevalence of artifactual reads with primer 1 as well as its reverse complement in the libraries generated in the current as well as previously published studies. Note the lack of such artifactual reads in the library generated from direct RNA sequencing. **C.** Genome browser track showing the alignment of an artifactual read twice at the same locus (*CNIH3*) in the opposite directions. Note that this is the only read that mapped to this locus; depth of coverage by minimap2 alignment is doubled in the places where the read maps twice. **D.** Secondary structure prediction of the read in panel **C** by RNA-fold shows that it is largely palindromic. **E.** Meta mountain plot of the extent of palindromes in the artifactual (orange, n = 50) and the same number of randomly selected (black) reads from the whole set (not restricted to non-semi-palindromic). Base pairing score of the given base from the RNA fold structure is increased by 1 if the base is paired downstream and decreased by 1 if the base is paired upstream; no change for an unpaired base. Scores were scaled to 100 bp meta-transcript. Shaded region indicates SD. **F.** Measurement of the coverage (left) and abundance (TPM; right) of transcripts by StringTie before and after filtering the supplementary alignments. **G.** Model depicting a possible source of artifactual reads in the long-read libraries. Primer 1 initiates synthesis of the first strand by the reverse transcriptase (RT). A micro-homology region in a read may create a short hairpin which would lead to switching of the template strand from the original RNA molecule to the first strand which is being synthesized, producing a continuous palindromic read.

Mix also assembled many incomplete ERCC transcripts where the assembled transcripts were shorter than the actual molecular standards. Most importantly, TASSEL, by far, assembled the highest number of complete ERCC transcripts (S5B Fig). When tested as a function of transcript abundance, we observed a large variation in the rate of false negative assembly of ERCC transcripts amongst the tested assembly methods (Fig 5C). TASSEL showed a substantially lower false negative rate than other methods, indicating its higher sensitivity. As alternative TSS and TTS of transcripts dictate most isoform differences across human tissues [72], a key parameter of assembling the correct and coherent transcript is determination of its ends. The

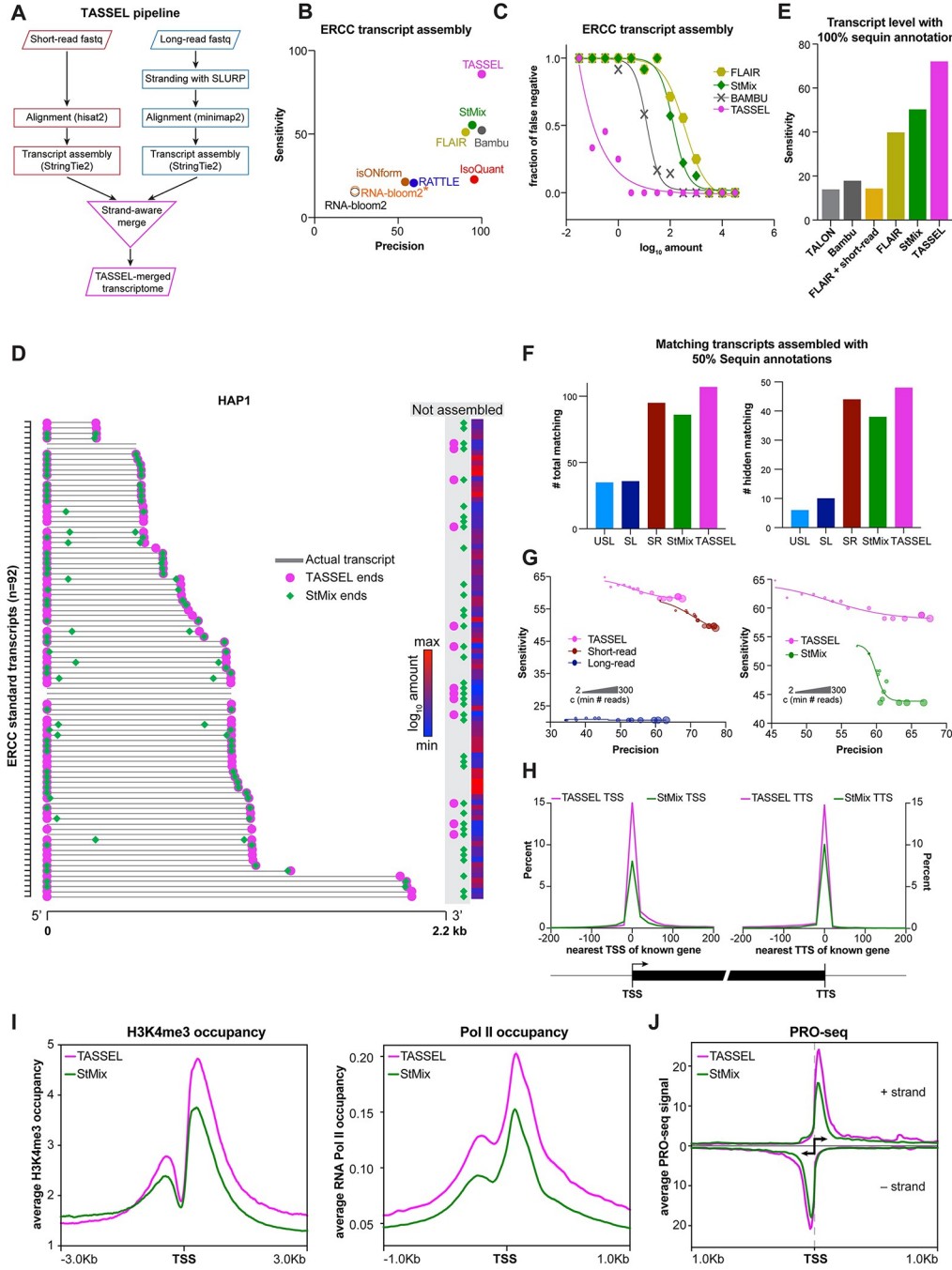

**Fig 5. Improved transcript assembly with a strand-aware hybrid pipeline. A.** Workflow of TASSEL (Transcript Assembly using Short and Strand-Emended Long reads) pipeline. Transcripts obtained from short-read RNA-seq are merged with those obtained from stranded long reads in a strand-aware manner. **B.** Comparison of the indicated transcript assembly methods for the sensitivity and precision of ERCC transcript assembly in the HAP1 dataset. RNA-bloom2* refers to transcript assembly by RNA-bloom2 using correction from the corresponding short reads. **C.** False negative rate of assembling ERCC transcripts, by each of the indicated assembly methods, as a function of their abundance in the HAP1 dataset. Each point represents only the ERCC transcripts at the given concentration. SLURP-stranded reads were used for FLAIR assembly. **D.** Ends of the 92 ERCC transcripts (arranged in the increasing order of length) assembled by TASSEL (magenta circle) and StringTie Mix (StMix, green diamond) in the HAP1 dataset. Gray bar indicates actual transcript. The color bar indicates the abundance of the given transcript. **E.** Sensitivity of the indicated assembly methods at the transcript level for the sequin spike-in standards. 100% of sequin annotations were provided to each of the methods during assembly of transcripts. FLAIR + short-read indicates integration of the corresponding short-read dataset at the FLAIR correction step. SLURP-stranded reads were used for FLAIR assembly.

Assembled transcripts were then compared against the sequin reference annotation using gffcompare. **F.** Number of total (*left*) or hidden (*right*) matching transcripts assembled by the indicated assembly method when only 50% of the sequin annotations were provided during transcript assembly. Hidden transcripts refer to the sequin standard transcripts whose annotations were removed from the annotation file. USL: unstranded long-read, SL: stranded long-read, SR: short-read, StMix: StringTie Mix. **G.** Sensitivity and precision of the indicated assembly methods under increasing stringency of transcript detection using 50% of the sequin annotations. Stringency was increased by increasing the minimum read counts (-c) required for calling a transcript; indicated by size of the marker. **H.** Proximity of TSS (left) and TTS (right) of known genes (gencode hg38v41) to the TSS and TTS of transcripts assembled by TASSEL or StringTie Mix in the HAP1 dataset. **I.** Enrichment of H3K4me3 (left, normalized to input) and RNA Pol II (right, normalized to the total number of mapped reads) at the TSS of transcripts assembled by TASSEL or StringTie Mix in the HAP1 dataset. H3K4me3 and RNA Pol II occupancy were calculated from ChIP-seq data from HAP1 cells [97,103]. **J.** Average PRO-seq signal at TSS of transcripts assembled by TASSEL or StringTie Mix on the positive (top) and negative (bottom) strands. Normalized PRO-seq data in HAP1 cells were obtained from [99].

ends of TASSEL-derived ERCC transcripts matched completely with the actual ends of these standard transcripts (Figs 5D and S5A). On the other hand, many transcripts assembled with StringTie Mix were artifactually truncated at their 5' and/or 3' ends. When tested on more complex sequin spike-in RNA standards, TASSEL showed highest sensitivity of transcript assembly when compared to TALON [73], Bambu, FLAIR and StringTie Mix (Fig 5E). To test the accuracy of *de novo* assembly capabilities, we tested TASSEL by randomly removing 50% of sequin annotations. TASSEL assembled the highest number of correct "total" and "hidden" transcripts when compared to the unstranded long-read, stranded long-read, short-read and StringTie Mix assemblies and TAMA pipeline [74] (Figs 5F and S5C). We observed that TAMA is highly reliant on reference annotations as it was able to assemble only one matching sequin transcript (compared to 107 for TASSEL) in the absence of reference annotations at the merge stage, and assembled only one matching "hidden" sequin transcript (compared to 48 for TASSEL) when 50% of sequin annotations were provided at the merge stage (S5C Fig). Expanding the scope of TASSEL, we observed a similar increase in the sensitivity of transcript assembly from datasets obtained from the PCR-cDNA kit (S5D Fig). Comparing the precision of transcript assembly, we observed that the baseline precision of TASSEL was lower than the short-read or StringTie Mix assembly (Fig 5G). However, the precision of TASSEL assembly can be increased to the equivalent levels of short-read assembly and StringTie Mix by increasing the stringency of minimum read counts required for transcript assembly. Importantly, TASSEL maintains higher sensitivity of assembly than other assembly methods at higher stringency of detection.

Next, we compared the entire transcriptome assembled in our HAP1 dataset by these programs with the reference transcripts. Although Bambu performed better than StringTie Mix for ERCC assembly, it had the lowest sensitivity of assembly at the transcript level (S5E Fig). Here again, TASSEL outperformed the other tested assembly programs. As StringTie Mix was closest to TASSEL in performance, we performed further comparisons between the two. In comparison to StringTie Mix, TASSEL showed a modest increase in transcripts that match completely to reference transcripts (S5F Fig, left). Importantly, there was a substantial decrease in the assembled transcripts that are contained within the reference transcripts (S5F Fig, right). Notably, the incidence of such intron retention transcripts is much lower in the TASSEL assemblies than those derived from only long reads. Based on our earlier analysis (Fig 2H), we rationalize that this improvement in TASSEL could be due to the incorporation of higher depth of short-read sequencing.

We also tested how TASSEL compares to StringTie Mix in the accurate determination of transcription start sites (TSS) and transcription termination sites (TTS) of all assembled transcripts. For this, we compared the TSS and TTS of TASSEL and StringTie Mix transcripts with those of known human genes (Fig 5H). TASSEL showed much better overlap with known TSS

and TTS than StringTie Mix, indicating that TASSEL assembles more coherent transcripts than StringTie Mix. To further evaluate the accuracy of the assembled 5'-ends, agnostic of the reference assemblies that may not reflect TSS in a particular lineage [72], we sought to compare TASSEL and StringTie Mix for enrichment of other genomic features of 5' ends from the same cell line. At the molecular level, transcription start sites are sites of high H3K4me3 deposition (epigenetic mark of active TSS [75]) and RNA Polymerase II (RNA Pol II) occupancy [76]. TSS derived from TASSEL were more enriched for these two functional entities than those derived from StringTie Mix (Fig 5I). Precision Run-On sequencing (PRO-seq), used to map the location of active RNA polymerase, can provide estimates of the 5' end of transcripts [77]. TASSEL-derived TSS showed substantially higher PRO-seq signal than StringTie Mix-derived TSS (Fig 5J), suggesting that TASSEL TSS are better indicators of active TSS than StringTie Mix TSS.

To fully characterize the capabilities of TASSEL, we then dissected the contribution of the individual components of TASSEL. First, we observed that the omission of SLURP-mediated stranding in TASSEL leads to an increase in the incorrect assembly of transcripts on the wrong strand and a decrease in the proportion of completely matching transcripts (S5G Fig), suggesting an important role of SLURP stranding in the overall efficacy of TASSEL. Then we investigated the contribution of short- and long-read transcripts to the TASSEL transcriptome (S5H Fig). We observed that both short- and long-read transcriptomes contribute equivalently to the TASSEL transcriptome. Furthermore, the consensus transcripts in the TASSEL transcriptome evince better enrichment of TSS and TTS (S5I Fig) as well as H3K4me3 occupancy, Pol II occupancy and PRO-seq signal (S5J Fig) than the transcripts obtained from either of the short- or long-read transcriptomes alone. Notably, TASSEL-mediated improvement is not limited to transcript ends. For example, TASSEL fully assembled a known spliced human transcript where the original short- or long-read dataset failed to do so (S5K Fig). As a proof of principle, we tested TASSEL on a PacBio [78] long-read dataset and observed that TASSEL leads to substantial improvement in assembly of the sequin spike-in standards (S5L Fig). We note that the HL1, HAP1, and sequin datasets tested here encompass a wide range of relative short to long read depth (S5M Fig). Efficacy of TASSEL on these datasets suggests its broad applicability on datasets of varying depths.

Collectively, our data show that TASSEL outperforms contemporary transcript assembly methods for correct and complete assembly of mono- and multi-exonic transcripts as well as enrichment of transcriptionally relevant features. Thus, we felt confident moving forward to use our method to interrogate assembly of cheRNA as a challenging testbed.

## TASSEL improves cheRNA identification

Based on the foregoing analyses, we reasoned that TASSEL can be used to improve the assembly of cheRNA which have been challenging to characterize due to their non-canonical features, low abundance, and high variance. Owing to its higher depth and dynamic range, we used only short-read RNA-seq of the chromatin and nucleoplasm fractions to define cheRNA. We defined cheRNA based on marked enrichment (>4 fold) of a transcript in the chromatin fraction relative to the nucleoplasm fraction (Fig 6A). The control ERCC spike-in standards were detected at equivalent levels in the two fractions (S6A Fig). Using differential expression analyses, we detected 6,746 and 4,969 cheRNA genes in HAP1 and HL1 cells, respectively. The detected cheRNA genes were proximal to cell line-specific protein coding genes (S6B Fig), suggesting potential biological significance.

The exact transcript structure of the detected cheRNA is not clear due to caveats in short-read RNA-seq, such as lower coverage of transcript start and end sites (Figs 1F, 1G and S1H)

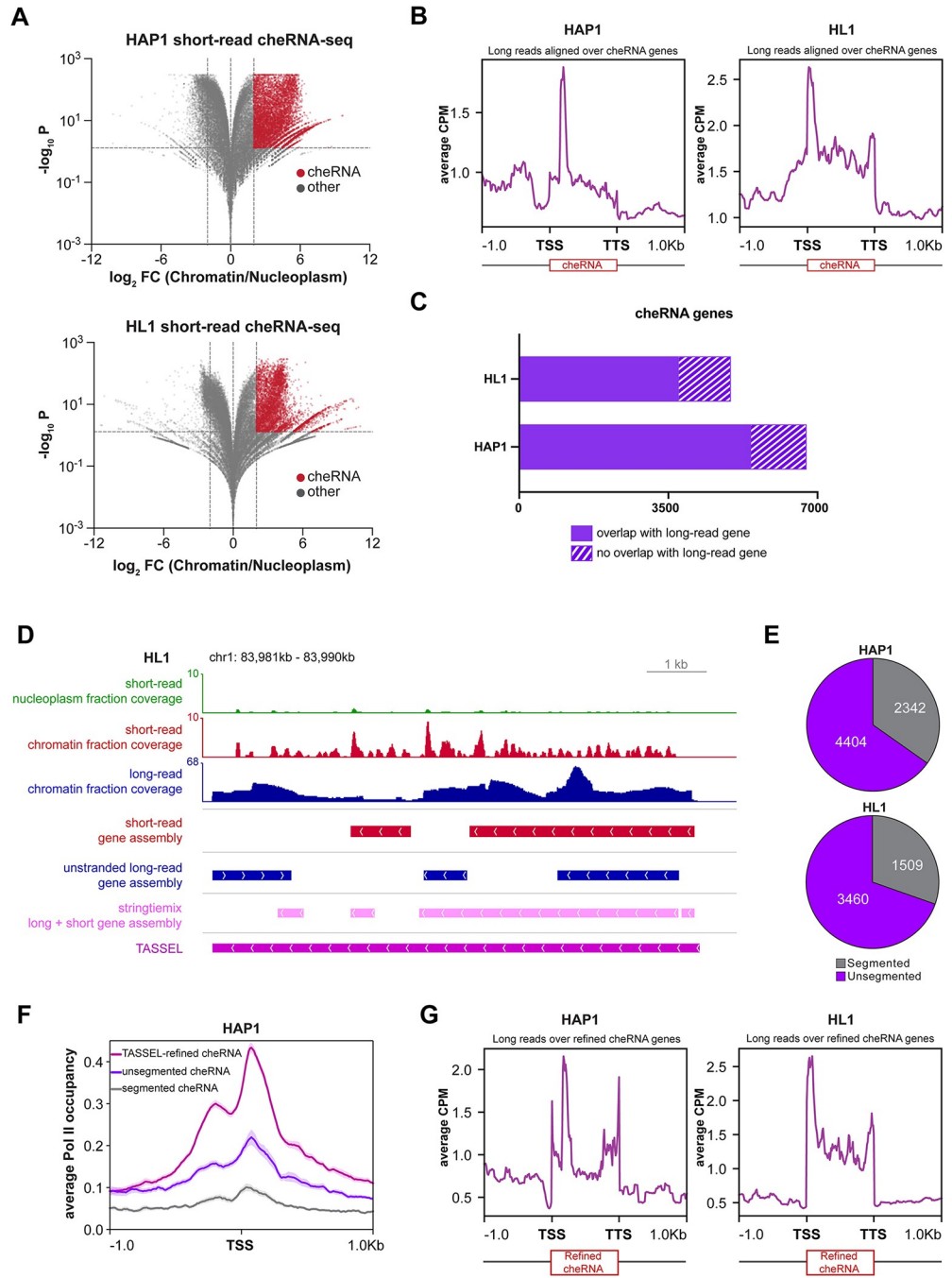

**Fig 6. Improved cheRNA characterization by integration of short-read assemblies with stranded long-read assemblies. A.** Volcano plot for enrichment of genes in the chromatin fraction vs adjusted *p* value. DESeq2 on short-read counts by StringTie was used to obtain cheRNA genes (red) that are significantly (Benjamini and Hochberg adjusted $p < 0.05$) enriched (>4 fold) in the chromatin fraction as compared to the nucleoplasm fraction. **B.** Metagene plots and heatmaps of mapped read coverage (average CPM) for HAP1 (left) and HL1 (right) long-read alignments, scaled to the cheRNA genes in the corresponding samples. **C.** Number of cheRNA genes with or without any overlapping gene in the corresponding long-read datasets in HAP1 and HL1 samples. **D.** Example cheRNA locus depicting the efficacy of the optimized methodology and improved assembly of cheRNA genes. Top: coverage from short-read nucleoplasm and chromatin fractions shows high enrichment in the chromatin fraction (data presented are the sum of replicates). Bottom: assembly by indicated methods shows resolution of the segmentation problem by merging short-read with the stranded long-read assembly. **E.** The number of segmented cheRNA genes identified in HAP1 and HL1 short-read datasets after merging of short-read and stranded long-read transcriptomes. **F.** Metagene plot depicting average RNA Pol II occupancy (solid lines; shaded region indicates SE) at the TSS (± 1 kb) of segmented,

unsegmented and TASSEL-refined cheRNA genes in the HAP1 dataset. RNA Pol II occupancy was calculated from ChIP-seq data from HAP1 cells [97]. **G.** Metagene plots of mapped read coverage (average CPM) for HAP1 (left) and HL1 (right) long-read alignments, scaled to the TASSEL-refined cheRNA genes.

and apparent segmentation of transcripts due to the short length of aligned reads. When compared to the long-read dataset, we observed moderate to high enrichment of long read coverage across cheRNA genes in comparison to their upstream and downstream regions (Fig 6B), suggesting that cheRNA genes, identified through short-read RNA-seq, are represented in the long-read datasets. Then we directly compared the cheRNA genes from the short-read datasets with the transcriptome assembled from the long-read datasets. More than 75% of cheRNA genes from the short-read dataset had an overlapping gene in the corresponding long-read dataset (Fig 6C). At a finer scale, more than 40% of cheRNA genes from the HAP1 short-read dataset and more than 20% from the HL1 short-read dataset had a counterpart in the long-read datasets with more than 90% overlap (S6C Fig). We observed that cheRNA genes with minimum overlap were significantly less abundant than those with maximum overlap (S6D Fig). While this analysis shows the potential of long-read gene assembly in identifying cheRNA gene structures inferred from short-read analysis, it also reveals that long reads fail to assemble coherent cheRNA transcripts in regions of low coverage. In addition to low coverage, other inherent platform-specific variations (Fig 2C, 2G and 2H) could also contribute to incongruence of short-read cheRNA genes. An example is highlighted in Fig 6D, where a cheRNA locus is detected on the basis of high enrichment in the chromatin fraction as compared to the nucleoplasm fraction in the short-read data. Although the coverage track indicates a single gene, StringTie assembly of short reads predicts two cheRNA genes at this locus. The StringTie assembly of the original long reads resulted in assembly of three transcripts–one on the Crick strand and two on the Watson strand. StringTie Mix [29] also fails to predict the correct transcript. Strikingly, use of TASSEL led to the assembly of one transcript in the correct direction. TASSEL not only assembled the transcript on the correct strand but also resolved the segmentation problem. We computed that ~30% of the original cheRNA were segmented prior to application of TASSEL (Fig 6E). Importantly, both mono and multi-exonic segmented cheRNA are refined by TASSEL (S6E Fig). To test that this computational approach is biologically meaningful, we compared RNA Pol II occupancy at the TSS of originally segmented or unsegmented cheRNA genes with TASSEL-refined cheRNA genes. Consistent with the expectation of Pol II enrichment decorating actively transcribed promoter regions [76], we detected a substantial increase in Pol II enrichment at the TSS of refined cheRNA (Figs 6F and S6F). There were also better indications of divergent transcription–a characteristic of *bona fide* promoters–from the TSS of refined cheRNA genes. Additionally, there was marked improvement in coverage of long reads over refined cheRNA genes with sharpened demarcation of 5' and 3' ends (Fig 6G vs 6B), suggesting convergence of the two datasets for better delineation of transcripts. These analyses highlight the utility of TASSEL for improved assembly of correct transcripts and genes, even for one of the most challenging classes of molecules.

## Discussion

With the increased interest in long-read RNA-seq, current efforts are directed towards weighing the merits of long-read RNA-seq over its short-read counterpart as well as development of new tools for improved coalescence of the two platforms [20,29,48,79,80]. To examine whether short- or long-read RNA-seq data or a combination of both would permit robust transcript characterization, we compared key features of read alignment, read coverage, transcript assembly, and capture of biologically relevant attributes of transcripts from short- and long-read

RNA-seq. To sharpen the challenge, we focused on chromatin-enriched RNA, a class of largely unannotated, low abundance transcripts that have presented problems for previous short read analyses [37,38]. We performed Illumina short-read and ONT direct cDNA long-read RNA-seq in parallel on multiple biological replicates in cell lines of mouse and human origin. Consistent with the performance of the two platforms, we obtained 2-6X coverage/bp for short-read sequencing and 0.5–0.7X for long-read sequencing. Alignments of reads from both platforms to the corresponding genomes were highly correlated between replicates, attesting to their reproducibility. Use of mono-exonic spike-in ERCC [44] and multi-exonic sequin [48,56] RNA standards enabled us to objectively compare the performance of the two platforms. Both showed a near-linear relationship for observed versus expected transcript abundance, validating their use in transcript quantitation and differential gene expression analyses. Owing to its higher sequencing depth, the short-read platform had a greater dynamic range for transcript detection, fraction of transcript covered as well as quantitation. In contrast, long-read coverage was more homogenous across ERCC transcripts and provided better coverage of 5' and 3' ends than short-read alignments. Collectively, our data suggest that short-read RNA-seq has quantitative advantages whereas the long-read RNA-seq has enhanced qualitative attributes. A hybrid approach that retains the best aspects of each can result in more definitive transcriptome characterization.

## Strengths and weaknesses of short- and long-read RNA-seq

As transcript assembly is critical for most of the downstream analyses, we systematically compared multiple transcript assembly approaches. Consistent with previous studies [53,54], reference-guided assembly with StringTie2 showed higher sensitivity as well as precision for transcript assembly for both short- and long-read RNA-seq, relative to other tested programs. A large fraction of long-read transcripts was fully contained within the reference intron, perhaps largely due to the enrichment of nascent transcripts in chromatin fractions of nuclear RNA [59–61] or limitations of long-read sequencing such as low coverage and internal priming events of oligo-dT based primers. We observed a high overlap between transcripts assembled in the short- and long-read datasets. A direct comparison of the structure of transcripts showed that many in the long-read datasets were "contained" in the short-read transcripts, largely due to the higher depth of the short-read sequencing. Therefore, multiple limitations of long-read sequencing can cause truncation of assembled transcripts at low coverage regions. The lower depth of long-read assembly also impacted the quantitation of low abundance transcripts when compared to the short-read transcriptome, cautioning its use for pan-transcriptome quantitation on its own.

## SLURP strands and improves transcript assembly

The lack of strand of origin information in the ONT direct and PCR-cDNA long-read sequencing hinders direct comparison to reference transcriptomes and short-read assemblies. Consistent with this limitation, we report clear evidence of mapping of ~40% of long reads to the incorrect strand and consequent ambiguous assembly of transcripts. The existing stranding methods discard a majority of reads leading to loss of information critical for high fidelity transcript assembly. To address this problem, we developed and benchmarked an accessible computational stranding pipeline, SLURP. Using SLURP, we corrected the erroneous mapping of long reads, resulting in a striking improvement in the assembly of spiked-in ERCC and sequin transcripts as well as assembly of transcriptomes from human and mouse cell line datasets. SLURP outperforms other stranding pipelines such as UNAGI and Pychopper, perhaps due to the way the sequences are searched within the reads and/or tolerance of mismatches.

Importantly, SLURP stranding is effective in improving the assembly of mono-exonic as well as complex multi-exonic transcripts with spliced isoforms. Although SLURP strands 60–75% reads, the loss of remaining unstranded reads does not seem to hamper the sensitivity of transcript detection or assembly. On the contrary, SLURP stranding leads to an overall increase in the sensitivity and precision of transcript assembly (Figs 3H and S3L) due to the removal of the underlying noise caused by misaligned reads that otherwise reduces the confidence of mapping in the correct orientation and creates discontinuity in intact transcript assembly (Figs 3I and S3O). StringTie Mix [29], a hybrid assembly program, permitted comparable stranding efficiency but resulted in truncated transcripts, seemingly due to overweighting of short reads in the hybrid assembly. Unlike StringTie Mix, SLURP can be used as a general-purpose standalone pipeline without the corresponding short-read data and independent of the assembly program. Importantly, our stranding pipeline can be broadly applied to preexisting long-read RNA-seq studies for improved transcript assembly, comparison and integration with short-read RNA-seq data, and high-confidence detection of *bona fide* antisense transcripts shown to play a critical role in gene regulation [81].

## Prevalent artifacts in direct ONT cDNA long-read sequencing

Our in-depth stranding analysis led to the serendipitous discovery that long-read ONT cDNA libraries are plagued with a cDNA synthesis artifact. Surprisingly, ~6% of long-read cDNA libraries are palindromes (within base-calling error). As we did not detect such reads in the ONT direct RNA library, we attribute these reads to an artifact of first strand synthesis, wherein reverse transcriptase switches from its initial RNA template to its product cDNA strand for further elongation. This finding is distinct from previously reported reverse transcriptase artifacts such as artificial splicing, in which the reverse transcriptase jumps from one direct repeat to another direct repeat of the same RNA or different RNA molecule, causing the deletion of the intervening sequence [82,83]. There is less chance of formation and detection of such palindromes in the short-read datasets as short-read libraries are synthesized from fragmented RNA and sequenced with shorter read lengths, often with enzymes that lack template switching activity. Alignment using standard long-read aligners such as minimap2 leads to double mapping of these reads to the same genomic region and flagging of one of the two alignments as a supplementary alignment. However, such an alignment not only inflates coverage statistics at the alignment stage but also leads to imprecise assembly and quantitation of all transcripts. It is important for the field to be aware of this likely prevalent artifact in long-read cDNA libraries which, to our knowledge, has not yet been reported. In principle, alternate cDNA library preparations with non-template switching reverse transcriptases or future aligners and assembly algorithms adapted to resolve these palindromes can be employed to accommodate these pitfalls.

## Enhanced detection of molecular features promises to improve functional analyses

Alternative transcription start and termination sites are shown to be the major drivers of isoform variance of transcripts [72]. Knowledge of transcript ends is also critical for downstream experimental manipulations to evaluate function. For example, efficiency of CRISPRi [84] targeting drops off sharply as a function of distance from the TSS [85]. Moreover, effective interpretation of experiments involving insertion of a strong polyadenylation sequence to make an RNA-level knockout [33,86] relies on accurate knowledge of the TSS to minimize residual RNA fragment size. In our study, direct comparison of short- and long-read alignments over known genes and curated transcription start and termination sites exposed informative

differences between the two platforms. The ends of the transcripts are much better represented in the long-read alignments than in the short-read alignments. We reason that this difference may arise due to multiple factors such as incomplete cDNA synthesis from random hexamer primers on fragmented RNA and heterogenous mapping of segmented short reads. In contrast, full-length cDNA synthesis on unfragmented RNA using a 3'-based oligoT primer and potential sequencing of the entire cDNA culminate in end-to-end coverage of transcripts in the long-read data set. This provides a qualitative advantage to the long-read RNA-seq for full-length characterization of transcripts.

## TASSEL improves integration of short- and long-read transcript assembly

Informed by the strengths and caveats of each of the two platforms, we developed a hybrid pipeline, TASSEL, to merge the short-read transcriptome with the stranded long-read transcriptome. When tested on ERCC standards against commonly used long-read transcript assembly methods such as FLAIR [55], Bambu [67], StringTie Mix [29], RNA-Bloom2 [68], IsONform [69], IsoQuant [70] and RATTLE [71], TASSEL outperformed other methods by assembling the highest number of matching spike-in transcripts. We further tested TASSEL on complex multi-exonic sequin standards and the human transcriptome in our HAP1 dataset. TASSEL had the lowest false negative rate of transcript assembly and unlike other methods, TASSEL led to assembly of complete transcripts on the correct strand. FLAIR was limited by low sensitivity and incomplete assembly. Although StringTie Mix was able to assemble transcripts on the correct strand, it had a substantially higher false negative rate than TASSEL and assembled many transcripts with truncated ends. We note that with lower stringency of detection, TASSEL may suffer from reduced precision. However, users can enhance the precision of TASSEL by increasing the stringency of detection while still achieving higher sensitivity than other methods tested here. Analyses based on spliced sequin spike-in standards as well as on HAP1 dataset indicate that TASSEL improves transcript assembly of complex multi-exonic transcripts with spliced isoforms. TASSEL also led to a substantial reduction in the transcripts that are assembled within an intron of the reference gene. We reason that this happens due to the amelioration of long-read artifacts such as low coverage and internal priming events of oligo-dT primers by short-read transcripts which are assembled from high depth libraries prepared without oligo-dT primers. Another reason for the higher efficacy of TASSEL is the use of stranded long reads which not only resolves the conflicting orientation of neighboring reads but also enhances the coverage of correctly assembled transcripts. Additionally, as TASSEL is employed after alignment and assembly of short- and long-read transcripts, it is not biased towards short-read transcript depth and maintains long-read information. As a test of its advantage at the functional level, TASSEL had higher enrichment of known TSS, TTS, histone modification of active TSS, and RNA Pol II than StringTie Mix. Together, TASSEL outperforms contemporary assembly methods for assembly on the correct strand, complete end-to-end assembly highest sensitivity of assembly, and unbiased integration of high depth short-read sequencing with the qualitative enrichment of long-read sequencing, which collectively culminates to superior manifestation of biologically relevant transcriptomic features.

In addition to the ONT long-read sequencing, another prevalent long-read sequencing platform is PacBio Iso-seq [78]. We note that the inherent read correction module of Iso-seq data analysis correctly strands the long reads by orienting them in the 5' to 3' direction. We show that TASSEL-based merge of a transcriptome derived from stranded PacBio long reads with that derived from corresponding short reads improves transcript detection and assembly.

We tested TASSEL on cheRNA transcripts which have been challenging to characterize due to their low abundance in cells, high variation from one cell type to another, and lack of

canonical attributes of coding transcripts [37,38]. In comparison to short- or long-read data alone or combined analysis with StringTie Mix, the use of TASSEL led to a marked improvement in the assembly of cheRNA. It corrected the segmentation of many mono- as well as multi-exonic cheRNA genes, resulting in enhanced definition of cheRNA transcription start sites as defined by RNA Pol II enrichment [76,87]. As the majority of cheRNA are lncRNA, our TASSEL-based assembly marks the first successful use of a hybrid assembly method in enhanced detection as well as assembly of lncRNA. It can therefore be used to resolve open questions about lncRNA such as their true molecular identity, convergence with other genomic as well transcriptomic features, and the role of antisense lncRNA. Collectively, our analyses of short- and long-read RNA-seq attributes and the development of stranding and merging pipelines can inform and improve future long-read transcriptome analyses beyond the cheRNA transcripts tested here.

## Materials and methods

### Cell culture

HAP1 cells were grown in IMDM media (Gibco, 12440–053), supplemented with 10% FBE (Seradigm, 3100–500) and 1% Pen/Strep (ThermoFisher 15140–122) to 80–90% confluency in T75 flasks at 37˚C. HL1 mouse cardiomyocytes (#SCC065) were grown per manufacturer instructions in supplemented Claycomb Medium (51800C-500ML) + 10% FBS (TMS-016) + 0.1mM Norepinephrine with 30mM L-ascorbic acid (A0937, A7506) + 5% L-Glutamine (G7513) in 0.1% gelatin-coated (SF008) T75 flasks.

To prepare the cells for RNA harvesting, they were washed once with 1X PBS, detached with TrypLE Express (Gibco #12605010), quenched with FBE-containing media, then washed with PBS, and pelleted by centrifugation for 5 min at 250 x g. Cell pellets were either snap frozen with liquid nitrogen or processed immediately.

### Nuclear fractionation and RNA isolation

Cell pellets were resuspended in Buffer A (10 mM HEPES•KOH pH 7.5, 10 mM KCl, 10% glycerol, 340 mM sucrose, 4mM MgCl$_2$, 1 mM DTT, 1 x Protease Inhibitor Cocktail (PIC) [1mM AEBSF, 0.8 μM aprotinin, 20 μM leupeptin, 15 μM pepstatin A, 40 μM bestatin, 15 μM E-64; from 200x DMSO stock]). Nuclear fractions were separated as described previously [37,38]. 1 μL of a 1:10 diluted ERCC RNA Spike-in standards aliquot (Life Technologies 4456740) was added to each nuclear fraction prior to addition of TRIzol reagent (Life Technologies 15596026). RNA from each fraction was extracted as described previously [37,38]. RNA was processed using the Zymo RNA Clean & Concentrator kit (Zymo Research R1017) with on-column DNase digestion. Ribosomal RNA was removed using the Ribo-zero Gold rRNA depletion kit (Illumina MRZG12324) for HL1 short-read RNA-seq samples and the RiboMinus kit (Invitrogen A15026) for HL1 long-read RNA-seq as well as HAP1 short- and long-read samples.

### Library preparation for short-read RNA-seq

100 ng of rRNA-depleted RNA from each nuclear fraction was used with the NEBNext Ultra II Directional library kit (NEB E7765S) to prepare RNA-seq libraries. RNA-seq libraries of nuclear fractions (chromatin pellet extract and soluble nuclear extract) from three independent cultures were sequenced through the University of Chicago Genomics Core Facility on the Illumina HiSeq 4000 to obtain 50 bp single end reads.

## Library prep for long-read RNA-seq

Two HAP1 and two HL1 long-read sequencing libraries were generated as follows. RNA from each fraction was extracted with Zymo RNA Clean & Concentrator (R1019) and subjected to qPCR analysis (primer sequences in S2 Table) for confirmation of enrichment for respective fractions (see S1A Fig). Next, the chromatin fraction RNA was depleted for ribosomal RNA with the RiboMinus Eukaryote Kit v2 (A15020) and precipitated with ethanol, then polyadenylated with E. coli Poly(A) Polymerase (M0276S) (for later priming with long read sequencing primers) and precipitated with ethanol. Successful rRNA depletion was assessed using BioAnalyzer. Finally, rRNA-depleted and polyadenylated chromatin RNA was subjected to library preparation according to the Oxford Nanopore direct cDNA sequencing kit protocol (SQK: DCS109). Libraries were checked for purity and size on a TapeStation with genomic DNA reagents (5067–5366) and tape (5067–5365). 10–50 fmol of library was loaded onto a MinION (MIN-101B) flow cell (R9.4.1; FLO-MIN106D) and run for 48 hours. Real-time basecalling information and FASTQ files were generated with the MinKNOW software using default settings (HAP1 rep1: minQ 7; HAP1 rep2: minQ 7; HL1 rep1: minQ 7; HL1 rep2: minQ 9).

## Alignment of short reads

Read quality and adapter content was checked using FastQC [88]. All short-read RNA seq libraries passed the quality check. Reads were then aligned to reference genomes (hg38 for HAP1 and mm10 for HL1 reads, catenated with ERCC sequences) using hisat2 version 2.1.0 [47] with—rna-strandness R—dta options. Sam files were converted to sorted bam files using samtools [89].

## Long read processing and ERCC analysis

FASTQ files generated by the MinKNOW software for each cell line were concatenated and mapped either to the hg38+ERCC (HAP1) or mm10+ERCC (HL1) genomes using minimap2 [46] without canonical splice sites (-ax splice -un). For stranding analyses, FASTQ files were subjected to our stranding pipeline prior to mapping (see Stranding methodology section below). For coverage analyses between replicates, HL1 replicate 1 was further filtered post-run to have a minimum phred score of 9 to match the minimum phred score of 9 used for replicate 2 using NanoFilt (-q 9). After sorting and indexing the BAM files (samtools), ERCC counts were computed using BEDTools coverage [58] and averaged between replicates; transcripts with count averages above 0 were kept. Then, coverage analyses were performed using Stringtie2 [54] with options -A to generate gene coverage in addition to transcripts and -L to specify long reads. Replicates were then analyzed in tandem using the Stringtie—merge function with expression estimation mode on (-e) to create a reference GTF with the union of transcripts between both replicates and subsequently re-estimate coverage for both genes and transcripts in individual replicates. Tags per million (TPM) was used for analysis of coverage. For the ERCC graphs, coverage was plotted against amount of ERCC added, length, and fraction of nucleotides mapped. Coverage of ERCCs was analyzed between long and short read datasets by first averaging ERCC coverage between replicates (excluding any transcripts with coverage of 0), and then plotting the averaged replicate coverages between long and short read datasets. Coverage analysis for all transcripts between long and short read datasets was performed by re-estimating coverage between the replicates with Stringtie2, then re-estimating coverage between long and short read re-estimated datasets and plotting the TPMs. As a supplementary comparison analysis, BEDTools counts vs StringTie TPMs were also plotted against each other for ERCCs.

### Datasets with sequin spike-in RNA standards

The short- and long-read dataset containing sequin spike-in RNA data were obtained from SG-Nex consortium [48]. We used data from three replicates each for direct cDNA, PCR-cDNA and Illumina short-read sequencing of HCT116 samples that were spiked with sequin RNA standards. The sequin dataset for PacBio platform was obtained from the GEO database (accession # GSE172421).

### Correlation of alignment of reads

For testing the correlation of alignment, bam files were converted to counts per million (CPM)-normalized bigwig files using the bamCoverage function of deepTools (-bs 1—normalizeUsing CPM) [90]. Then, the multiBigwigSummary function of deepTools was used to compute correlation of coverages of aligned reads at all genomic regions binned at 1 kb region (-bs 1000). Correlation between the replicates was then plotted using plotCorrelation function (-c pearson -p scatterplot—log1p).

### Metagene plots

To plot the aligned short and long reads on the known genes: Coordinates of the known genes (mm10vm23 assembly for mouse genome and hg38v41 assembly for human genome) were obtained from the UCSC table browser [91]. Counts per Million (CPM)-normalized bigwig files (binned at 1 base pair) of aligned reads were then used to score mean coverage over the known genes using the computeMatrix function of deepTools in scale-regions mode (-b 1000 -a 1000—missingDataAsZero) and plotted using plotHeatmap.

To plot the aligned short and long reads on transcription start sites and polyA sites: Coordinates of annotated transcription start sites were obtained from refTSS [50] and polyA sites from polyASite [51]. Mean coverage over these coordinates were then calculated using the reference-point mode of computeMatrix (-b 1000 -a 1000—missingDataAsZero) and plotted using plotHeatmap.

## Comparison of transcript assembly programs

For long-read transcript assembly, we compared FLAIR, StringTie *denovo* and StringTie guided approaches. For sequin transcript assembly, either the complete reference annotation or random 50% annotations were provided during the guided mode of transcript assembly for each of the programs tested. To assemble transcripts with FLAIR, sorted bam files were converted to bed12 format using the bam2Bed12.py script of FLAIR. FLAIR correct was used with reference genome fasta and annotation files (Figs 5E and S2A) or short-read gtf files (Fig 5E) (flair correct—nvrna -q input.bed -g refgenome.fa -f refgenome.gtf/short-read.gtf -o FLAIR-correct_output.bed). The corrected bed file was then used to obtain a gtf file using FLAIR collapse (flair collapse -g refgenome.fa -q FLAIR_corrected.bed -f refgenome.gtf/short-read.gtf -r sample.fq -o flair_collapse_output.gtf). For StringTie, sorted bam files for each sample were used to assemble transcripts using StringTie version 2.1.1 [54] with the -L option in either *de novo* or guided assembly mode (-G reference_annotation.gtf). For guided assembly, reference annotation gtf files of the gencode hg38v35 assembly for HAP1 samples and the mm10 assembly for HL1 samples were concatenated with gtf files for the ERCC spike-in RNA mix.

For short-read transcript assembly, sorted bam files for each sample were used to assemble transcripts using cufflinks version 2.2.1 (-u -N—library-type fr-firststrand) [92] or StringTie version 2.1.1 (—rf) [54] in either *de novo* or guided assembly mode. In cufflinks guided assembly mode, reference genome transcripts that are not detected in the sample are appended to

the sample gft files with their coverage marked as "0.0000". As it would artifactually inflate the sensitivity of assembly, we removed all transcripts with zero coverage in the sample gtf files.

To compare the efficacy of transcript assembly programs, gtf files of transcripts assembled by each of the programs was compared with corresponding reference genome gtf files using gffcompare [93]. Sensitivity (*True positive / (True positive + False negative)*) and Precision (*True positive / (True positive + False positive)*) of transcript assembly were compared to gauge the efficacy of assemblers.

### Testing stranding with UNAGI and Pychopper

UNAGI [65] was used in the default mode (-i long-read.fastq -g ref_genome.fa). It yields a fastq file with stranded reads. Number of reads in this stranded fastq file was used to calculate percent stranding.

Pychopper (https://github.com/epi2me-labs/pychopper) was run using the kit-specific primer configuration. Full length stranded reads yielded by Pychopper were then used to map to the reference genome using minimap2 as described above.

### SLURP (Stranding Long Unstranded Reads using Primers) stranding methodology

We used MEME analysis [64] to predict the sequences of primers used in the cDNA library prep. These primer sequences were then used as guides to ascertain the strand of origin of long reads. For this, the following stranding methodology was adapted:

1. Reads with partial primer 1 sequence in the first 100 bp of reads, permitting 2 mismatches, were extracted using:

seqkit grep -s -i -P -R 1:100 -m 2 -p GCTCTATCTTCTTT

2. Reads with the reverse complement of partial primer 2 in the last 100 bp, permitting 2 mismatches, were extracted using:

seqkit grep -s -i -P -R -100:-1 -m 2 -p CCCAGCAATATCAG.

3. Reads from 1 and 2 were combined and deduplicated.

4. Reads with partial primer 2 sequence in the first 100 bp of reads, permitting 2 mismatches, were extracted using:

seqkit grep -s -i -P -R 1:100 -m 2 -p CTGATATTGCTGGG.

5. Reverse complement of reads from (4) were made using:

seqkit seq -t dna -r -p.

6. Reads from (3) and (5) were combined and deduplicated to obtain stranded reads.

A bash script of this stranding pipeline is made available on at the GitHub repository (https://github.com/kainth-amoldeep/SLURP). We note that this default strategy will create a second strand library. If required, a first strand library can be created by obtaining the reverse complement of the SLURP output by using seqkit seq -t dna -r -p.

### Detection and analysis of the palindrome artifact in long-read sequencing

We extracted common reads which had primer 1 in their first 100 bp (seqkit grep -s -i -P -R 1:100 -p GCTCTATCTTCTTT) and the reverse complement of primer 1 in their last 100 bp (seqkit grep -s -i -P -R -100:-1 -p AAAGAAGATAGAGC).

These reads were then analyzed using RNAfold [94], as we reasoned that the secondary structure prediction of RNAfold could be used to detect potential palindromes in long reads. From the RNA-fold dot and bracket output (dot denotes unpaired residue and bracket denotes paired residue), we calculated the percentage of brackets in the given read to determine extent of residues base-paired in the long-reads. In addition, RNA-fold also provides minimum free

energy (MFE) scores of residues in the reads. We binned and scaled MFE scores of multiple reads to 100 bp to obtain a meta-mountain plot of MFE scores.

For mapping and counting statistics, the palindromic reads were aligned to the reference genome using minimap2 (-ax splice -un—MD). As palindromes were tagged with supplementary alignment to the same locus, we filtered out reads with the supplementary alignment sam tag using samtools (view -F 2048). The filtered and non-filtered bam files were then used to assemble and quantify transcripts using StringTie to compare coverage and abundance (TPM) of transcripts.

## Comparison of short- and long-read transcriptomes

For comparing abundance of transcripts and genes in short- and long-read samples, String-Tie-assembled gtf files of long-read samples (2 replicates) and short-read samples (3 replicates) were merged using stringtie—merge to obtain a non-redundant pool of a unified set of transcripts across samples. This merged transcriptome gtf file was then used as a reference for re-estimation of transcripts in each of the samples using stringtie -e -B -G stringtie_merge_file.gtf -C coverage_file -A gene_abundance_file. Then, average transcript and gene abundance (Tags per million, TPM) of every transcript and gene was calculated across the replicates for short-read and long-read samples. These average TPMs were compared to test the correlation between short-read sequencing and long-read sequencing.

To test the overlap between short- and long-read transcripts, bedtools [58] intersect was used in default mode to obtain the number of long-read transcript with any overlap in the short-read data and *vice versa*. To compare the transcript structure assembled by short-read and long-read assembly, gffcompare was used with the gtf file of long-read assembly as a reference transcriptome and the gtf file of short-read assembly as the query transcriptome. The relationship between the reference and query transcripts was ascertained by the "class code" output of gffcompare.

## Enrichment of transcript ends in short and long reads

CPM-normalized bedgraph files were made from sorted bam files of each sample using the bamCoverage function of deepTools (-bs 1—normalizeUsing CPM -of bedgraph). Then, sorted bedgraph files were used to calculate mean signal at the 100 bp region around the TSS or TTS of known genes using bedtools [58] (map -a TSS_100bp.bed -b CPMnormlized.bg -c 4 -o mean > bedtoolsmapped_TSS100bp_mean). The average signal thus obtained was then normalized to the average signal obtained from mapping reads to a similar number of random regions (100 bp wide) in the genome.

## Long-read gene fusion

Name-sorted bam files of each long-read sample were used as input to detect fusion events using LongGF [95] with <min-overlap-len> 100 <bin_size> 30 <min-map-len> 100. Then, grep "SumGF" LongGF.log > FusionList was used to get a list of detected fusions. The list of fusion events was then used to make a circos plot using BioCircos [96].

## TASSEL (Transcript Assembly using Short and Strand-Emended Long reads)

Long reads were stranded using SLURP (described above). Stranded long reads and short reads were aligned to the reference genome using minimap2 and hisat2, respectively. StringTie2 was

used to assemble transcripts in guided assembly mode (with -L option for long reads). Assembled transcripts were merged in a strand-aware manner using stringtie—merge—rf.

## Comparison of TASSEL with other assembly programs

As recommended for RATTLE [71], Pychopper-processed reads were clustered in isoform mode (./rattle cluster -i processed_reads.fq—iso -o rattle_cluster/), followed by RATTLE correction (./rattle correct -i processed_reads.fq -c rattle_cluster /clusters.out -o./rattle_correct/) and RATTLE polish (./rattle polish -i./ rattle_correct /consensi.fq -o rattle_polish/—summary). The output "transcriptome" reads were mapped using minimap to obtain a paf file. The transcript models were then extracted from the aligned paf file to make a gtf file, subsequently used with gffcompare for the evaluation of transcript assembly.

For IsoQuant [70], read alignments from minimap were used to assemble transcripts in the guided mode (isoquant.py—reference reference.fasta -g reference_annotat.gtf—complete_-genedb—bam aligned_sorted.bam—data_type nanopore—report_novel_unspliced true -o iso-quant_out/). The assembled transcripts in the output gtf file were then evaluated using gffcompare.

As recommended for For RNA-Bloom2 [68], Pychopper-processed reads were used in the default mode (java -jar RNA-Bloom.jar -long processed.fq -outdir RNA-bloom_out/). We also tested short read-based correction of RNA-bloom2 assembly (java -jar RNA-Bloom.jar -long processed.fq -sef short-read.fastq -outdir RNA-bloom_out/). The output "rnabloom.transcripts.fa" reads were mapped using minimap to obtain a paf file. The transcript models were then extracted from the aligned paf file to make a gtf file, subsequently used with gffcompare for the evaluation of transcript assembly.

As recommended for IsONform [69], Pychopper-processed reads were clustered in the ONT mode (isONclust—ont—fastq processed_reads.fq—outfolder./isONclust). The clustered fastq files were then made (isONclust write_fastq—N 1—clusters isONclust/final_clusters.tsv —fastq processed_reads.fq—outfolder isONclust/fastq_files), followed by correction using isONcorrect (run_isoncorrect—fastq_folder isONclust/fastq_files/—outfolder isONcorrect/ correction/). The transcript isoforms were then obtained using isONform (isONform_parallel. py—fastq_folder isONcorrect_output/—outfolder isONform_output/—split_wrt_batches). The output "transcriptome.fq" was mapped using minimap to obtain a paf file. The paf file was consolidated to make a gtf file, subsequently used with gffcompare for the evaluation of transcript assembly.

When comparing TASSEL to TALON [73], TAMA [74], FLAIR [55], Bambu [67] and String-Tie Mix [29], we used 100% sequin annotations, 50% sequin annotations or the corresponding mammalian annotations. For TALON, we initialized the database with the reference annotations (talon_initialize_database—f sequin_annotations.gtf—g talon_sequinannotation—a talon_sequi-nannotation—o talon_sequinannotation_database). Then, we labeled the reads to flag for internal priming (talon_label_reads—f sample_input.sam—g reference_fasta.fa—o talonLabelreads_sam-pleoutput). Then we did replicate-based filtering of transcripts (talon_filter_transcripts—db talon_sequinannotation_database.db -a talon_sequinannotation—o transcript_clean) followed by extraction of a gtf file for transcript models observed by TALON (talon_create_GTF—db talon_-sequinannotation_database.db -a talon_sequinannotation -b talon_sequinannotation—whitelist transcript_clean—observed -d dataset.csv—o talon_creategtf.gtf).

For TAMA, the minimap2 output of long read alignment was used for the TAMA collapse module using the recommended TAMA high methodology (python tama_collapse.py -d mer-ge_dup -x no_cap -a 300 -m 20 -z 300 -sj sj_priority -lde 3 -sjt 10 -s minimap2aligned_sam-pleinput.sam -f reference.fasta -p tama_high_sample). The output transcript models of

"TAMA collapse" for all replicates were then merged using the "TAMA merge" module (tama_merge.py -f list_tamacollapseoutput.txt -p tamamerged). The merged bed file was converted to a gtf file for comparison to the reference annotation. For assembly with FLAIR, sorted bam files of combined replicates (stranded with SLURP in Figs 5B, 5C, 5E, S5B and S5E) were converted to bed12 format using the bam2Bed12.py script of FLAIR. FLAIR correct was used with reference genome fasta and annotation files or short-read gtf files (flair correct —nvrna -q input.bed -g refgenome.fa -f refgenome.gtf/short-read.gtf -o FLAIRcorrect_output. bed). The corrected bed file was then used to obtain a gtf file using FLAIR collapse (flair collapse -g refgenome.fa -q FLAIR_corrected.bed -f refgenome.gtf/short-read.gtf -r sample.fq -o flair_collapse_output.gtf).

Bambu (version 3.1.1) was obtained from github/goekelab. An annotation file for Bambu was prepared using BambuAnnotations < prepareAnnotations(ref_annotations.gtf). Long-read alignments of combined replicates were obtained from minimap2 and were used to make a summarized experiment object using longread_Bambu_se <- Bambu(reads = longread.bam, annotations = BambuAnnotations, genome = ref.fa). Transcripts assembled by Bambu were extracted using Bambu_constructedAnnotations = longread_Bambu_se [assays(longread_Bambu_se)$fullLengthCounts > 0].

For StringTie Mix, bam files of combined short-read replicates and long reads replicates were obtained using hisat2 and minimap2, respectively. Reference annotation gtf files of the gencode hg38v35 assembly for HAP1 samples and the mm10 assembly for HL1 samples were concatenated with gtf files for the ERCC spike-in RNA mix. These files were then used with StringTie Mix (stringtie—mix short-read.bam long-read.bam -o stringtiemix.gtf -C stringtie-mix -G reference.gtf).

Stringency of transcript detection was increased by increasing the minimum number of reads required to assemble the transcripts using the -c option of StringTie (2, 4, 6, 8, 10, 15, 20, 25, 50, 75, 100, 150, 200, 250, 300). 50% reduced sequin annotations were provided during transcript assembly for each replicate of short- and long-read dataset. Assembled transcripts for each replicate were merged for the given program using the StringTie merge function as described above for StringTie, StringTie Mix and TASSEL. These merged transcriptomes, at each setting of -c option, were then used to test the sensitivity and precision of transcript assembly by comparing against the full set of sequin annotations using gffcompare.

For PacBio data, subreads of a PacBio long-read run and short reads of an Illumina NextSeq 500 run for H1975 cell line, spiked-in sequin standards, were obtained from GSE172421. Long-reads were aligned with pbmm2 and subjected to assembly using the StringTie -L option. Short-reads were aligned and assembled as described above using hisat2 and StringTie. The transcriptome obtained from each platform was TASSEL merged and compared to sequin annotations using gffcompare.

## Enrichment of H3K4me3, RNA Pol II and PRO-seq at TSS

Fastq files for H3K4me3 ChIP-seq in HAP1 cells (Input: SRR6671609, H3K4me3: SRR6671589) and RNA Pol II ChIP-seq for HAP1 cells (SRR2301045) [97] were downloaded from the GEO database. Reads were aligned to genome (hg38) using bowtie2 [98]. For H3K4me3, an input normalized bigwig file was made using the bamCompare function of deepTools (bamCompare -bs 1 -b1 H3K4me3_IP.bam -b2 Input.bam–operation ratio). For Pol II ChIP-seq, sorted bam files were directly converted to counts-normalized bigwig files using the bamCoverage function of deepTools (-normalizeUsing CPM). The bigwig files were then used to plot metagene profiles on TSSs of transcripts from StringTie Mix and TASSEL genes using the computeMatrix and plotProfile functions of deepTools.

For PRO-seq, bigwig files of normalized PRO-seq signal in HAP1 cells were obtained from GSM5829596 [99]. Forward PRO-seq signals were contoured over plus strand TSS and reverse PRO-seq signals were contoured over negative strand TSS of TASSEL or StringTie Mix transcripts using the computeMatrix function and plotted using the plotProfile function of deepTools.

## Estimation of cheRNA

StringTie and DESeq2 [100] were used to detect transcripts and genes which were significantly enriched in the chromatin fraction as compared to the nucleoplasm fraction in short-read sequencing. For this, gtf files of chromatin fraction reads and nucleoplasm fraction reads (three biological replicates each) were obtained by StringTie guided assembly of aligned reads. The transcripts from chromatin and nuclear fractions (as assembled by StringTie on the respective alignment files) are merged with default settings of StringTie—merge. Therefore, the use of a unified set of transcripts created by StringTie—merge to calculate ratiometric enrichment avoids the influence of sample-to-sample background heterogeneity as all the samples are re-evaluated against a common template. These gtf files were merged using default settings of stringtie—merge—rf to create a pool of high-confidence unified non-redundant transcripts across the samples for differential gene expression of known as well as *de novo* transcripts across experimental conditions. StringTie—merge also rectifies any incomplete transcript in a given sample if it detects a corresponding complete transcript in another sample. This unified and consolidated pool of transcripts across chromatin and nuclear fractions was then used as a reference transcriptome to re-estimate transcript structure and abundance in each of the replicates of both fractions using stringtie -e -B -G stringtie_merge.gtf. Raw counts of the re-estimated transcript and gene abundance were obtained using the prepDE.py script from StringTie. The raw counts were then used to calculate differentially expressed transcripts and genes in the chromatin vs nucleoplasm fractions using DESeq2. Transcripts and genes which were enriched more than four-fold in the chromatin fraction at adjusted $p < 0.05$ and had a length of more than 200 bp were deemed as cheRNA.

## Gene ontology for cheRNA proximal genes

Protein coding genes closest to cheRNA genes were computed using the closestBed function of bedtools (-a cheRNA_genes -b refgenome_proteincodinggene). The identified genes were then uploaded to the DAVID gene ontology portal [101] to calculate enriched biological processes categories.

## Comparison and merge of cheRNA with long-read sequencing

To plot long-read alignments on cheRNA genes: CPM-normalized bigwig files of long-read alignments were used to calculate mean coverage over cheRNA genes using the computeMatrix function of deepTools in scale-regions mode (-b 1000 -a 1000—missingDataAsZero) and plotted using plotHeatmap.

To calculate the extent of segmentation of cheRNA genes, genes in the TASSEL-merged gtf file were further consolidated using bedtools merge (-s -c 4,6 -o distinct). Then we intersected these consolidated genes with cheRNA genes using intersectBed (-wa -wb -s) to obtain the number of consolidated genes that intersected more than once with different cheRNA genes, essentially counting the segmented cheRNA genes.

## Pol II occupancy at cheRNA

Fastq files for RNA Pol II ChIP-seq for HAP1 cells (SRR2301045) [97] and eight-week old mouse heart (RNA Pol II IP: SRR489670; Input: SRR489681) [102] were downloaded from the

GEO database. Reads were aligned to respective genomes (hg38/mm10) using bowtie2 [98]. For HAP1 Pol II ChIP-seq, sorted bam files were directly converted to bigwig files using the bamCoverage function of deepTools. For mouse Pol II, an input normalized bigwig file was made using the bamCompare function of deepTools (bamCompare -bs 1 -b1 Pol_IP.bam -b2 Input.bam—operation ratio). The bigwig files were then used to plot metagene profiles on TSSs of cheRNA genes using the plotProfile function.

## Supporting information

**S1 Fig. Supporting data for Fig 1. A.** RT-qPCR confirmation of RNA fractionation into chromatin (purple) and nucleoplasm (green) extracts in the indicated samples. Protein coding transcripts (*GAPDH* and *B2M*) were used as representative of nucleoplasm enrichment while lncRNA (*MALAT1* and *PVT1*) were used as controls for chromatin enrichment. Error bar indicates SD for replicates (n = 3 for short-read; n = 2 for long-read). **B.** Density plot of read lengths from long-read sequencing of HL1 replicates (n = 2). **C.** Correlation between HL1 replicates (n = 2) for genomic coverage of long-read alignments binned at 1 kb. Pearson correlation is shown. **D.** Correlation between HAP1 (left) and HL1 (right) replicates (n = 3) for genomic coverage of short-read alignments (*top* nucleoplasm fraction; *bottom* chromatin fraction) binned at 1 kb. Pearson correlations are shown for each plot. **E.** Average counts per million (average CPM, dark color; +/- SD, lighter color) of aligned reads across all ERCC transcripts, meta-scaled to 1000 nt, in the HL1 short-read (n = 3) and long-read (n = 2) samples. **F.** Relationship between the average fraction of ERCC transcripts covered by short (left) or long (right) read alignments as a function of their amount (attomoles) or length (nt) for HL1 samples. **G.** Relationship between length and CPKM (coverage per kb per million mapped reads) of known genes in HAP1 short- and long-read datasets. Average CPKM for replicates are plotted against length of genes on a $\log_{10}$ scale. Colors correspond to kernel density estimations of scatter plot distribution. **H.** Metagene plots of mapped read coverage (average CPM) for HL1 short- and long-read alignments, scaled to transcription start sites (TSS) and transcription termination sites (TTS) of known genes. **I.** Metagene plot of mapped read coverage (average CPM) for HAP1 long-read alignments, scaled to transcription start sites (TSS) and transcription termination sites (TTS) of a subset of known genes which are similar in length distribution (0.2–2 kb) to ERCC genes. Note that the similarity of profile of this plot with that of ERCC genes (main Fig 1D, *right*) suggests that apparent gene body coverage decay in Fig 1F *right* is due to meta scaling of coverage on genes longer than the median long read. **J.** Metagene plots of mapped read coverage (average CPM) for HL1 short (top) and long (bottom) reads, centered on curated transcription start sites (TSS; [50]) or termination sites (polyA; [51]). **K.** Zoomed-in metagene plots of mapped read coverage (average CPM) for HAP1 short reads, centered on curated transcription start sites (TSS; [50]) or termination sites (polyA; [51]). Note the similarity of decay in signal in this plot to that observed for ERCC coverage (main Fig 1D, *left*). **L.** Circos plot depicting gene fusions supported by at least 2 reads in HAP1 (top) and HL1 (bottom) long-read datasets. Dark blue arcs indicate fusion events detected in both replicates; light blue arcs indicate fusion events detected in only one of the two replicates. The BCR::ABL fusion, a key genomic fusion, is highlighted in red. (PDF)

**S2 Fig. Supporting data for Fig 2. A.** Comparison of the indicated short-read (left) and long-read (right) transcript assembly methods for ERCC, sequin and HAP1 transcripts. Shown are the sensitivity and precision of assembly at the transcript level as computed by gffcompare. Note that apparent (app.) sensitivity and precision indicate comparison to reference human annotations which does not reflect ground truth for a given cell line. Error bar indicates SD

(n = 2–3). **B.** *Top*: Average abundance (Tags Per Million; TPM) of ERCC transcripts determined by StringTie versus amount added (attomoles) or length (nt) of the transcript in HL1 samples (n = 3 for short reads, n = 2 for long reads). *Bottom*: Average abundance (Tags Per Million; TPM) of sequin spike-in transcripts determined by StringTie versus amount added or length (nt) of the transcript in SGNex direct-cDNA samples of HCT116 samples (n = 3 for short and long reads). Short-read plots are shown in red (left), long-read plots in blue (right). $R^2$, correlation coefficient for linear regression. **C.** Comparison of abundance (TPM) of ERCC transcripts obtained from StringTie with BEDTools counts for HL1 long-read replicate 1. Both methods performed nearly identically ($R^2$ = 0.99, correlation coefficient for linear regression). **D.** Correlation between HL1 replicates for transcriptome assembly by StringTie. Shown are the scatter plots of abundance (TPM) of each transcript in the two replicates. Colors correspond to kernel density estimations of scatter plot distribution. $R^2$, correlation coefficient for linear regression. **E.** Comparison of structure of transcripts assembled by StringTie for HAP1 and HL1 short-read samples with structure of reference genome transcripts. The class codes for relationship between the assembled transcript and the closest reference transcript were deduced from gffcompare. **F.** Correlation plot for the average abundance (TPM) of ERCC spike-in transcripts in the short-read datasets (n = 3) with the average abundance in the long-read datasets (n = 2) of HL1 samples, determined by StringTie. Shown are the transcripts detected by both sequencing platforms. **G.** Scatter plot comparing the average abundance (TPM) of all transcripts in the short-read datasets (n = 3) and the long-read datasets (n = 2) in HL1 samples. Abundance was calculated using the re-estimation function of StringTie from the merged transcriptome. Color bars correspond to kernel density estimations of scatter plot distribution. **H.** Histograms of length of transcripts assembled in the short- and long-read transcriptome. The distribution is shown for transcripts up to 10 kb in length. **I.** Total number of transcripts assembled in the HL1 short-read dataset as a function of percent of short reads sampled. Dotted line indicates equivalent coverage compared to the corresponding long-read dataset in terms of total mapped bases.
(PDF)

**S3 Fig. Supporting data for Fig 3. A.** Percent of long reads stranded by UNAGI [65], Pychopper, and SLURP (Stranding Long Unstranded Reads using Primers) in the HAP1 and HL1 datasets. **B.** Detection of primer 1 and primer 2 sequences in the long reads using MEME [64] *de novo* motif discovery. **C.** Measurement of indicated stranding methods in terms of the number of transcripts assembled and extent of transcripts mapping to the wrong strand of the reference genome. 3crit indicates using primer 1, primer 2 and reverse complement of primer 2; 4crit indicates primer 1, reverse complement of primer 1, primer 2 and reverse complement of primer 2; m2 indicates allowing 2 mismatches and m3 indicates allowing 3 mismatches. **D.** Workflow of SLURP pipeline. Reads that contain the first-strand synthesis primer (primer 1) or reverse complement (rc) of the strand switching primer (primer 2) are non-redundantly merged with the reverse complement of reads that contain the strand switching primer. **E.** Comparison of Pychopper and SLURP for the precision of assembling the ERCC (left) or all (right) transcripts in the HAP1 dataset. **F.** Comparison of Pychopper and SLURP for the total number of matching ERCC (left) or all (right) transcripts in the HAP1 dataset. **G.** Example of an ERCC transcript correctly assembled using SLURP where Pychopper failed to do so. **H.** Comparison of Pychopper and SLURP for the sensitivity and precision of transcript assembly by FLAIR in the HAP1 dataset. **I.** Comparison of Pychopper and SLURP for the sensitivity and precision of transcript assembly, total number of matching transcripts, and the computational time required for the stranding of reads in the HL1 dataset. **J.** Same as I, for the HCT116 dataset from the SG-Nex consortium. **K.** Example of a sequin spike-in transcript annotation

with two spliced isoforms that vary in TSS and TTS, that is correctly assembled by SLURP-stranded long reads where the original unstranded reads failed to do so. **L.** Comparison of SLURP-mediated change in the sensitivity and precision of the sequin spike-in standards for the subset of splice variants with alternative TSS or TTS, e.g., the two transcript isoforms of *R2_20* in panel K. **M.** Efficacy of the stranding pipeline in a previously published dataset [48]. Shown are the percentage of transcripts mapping to the incorrect strand before and after SLURP-stranding. **N.** Efficacy of the SLURP-stranding pipeline upon retention or removal of unstranded long reads in the HAP1 dataset. Shown are the percentage of transcripts mapping to the incorrect strand. **O.** Example of a known gencode transcript assembled by SLURP-stranded long reads where the original unstranded reads failed to do so in the HAP1 dataset. (PDF)

**S4 Fig. Supporting data for Fig 4. A.** Bar plot of percentage of paired bases in the artifactual reads as a test for palindromes in the reads. RNAfold was used to determine paired bases in the secondary structure where a palindrome outweighs other structures. (PDF)

**S5 Fig. Supporting data for Fig 5. A.** Ends of the 92 ERCC transcripts (arranged in increasing order of length) assembled by TASSEL (magenta circle) and StringTie Mix (StMix, green diamond) in the HL1 dataset. Gray bar indicates actual transcript. The color bar indicates the abundance of the given transcript. **B.** Number and extent of ERCC standard transcripts assembled by the indicated assembly method in the HAP1 dataset. StMix: StringTie Mix. SLURP-stranded reads were used for FLAIR assembly. **C.** Number of matching sequin transcripts assembled by the indicated method when no reference sequin annotation (left) or 50% of the reference annotations (middle and right) were provided to TAMA at the merge stage. Note that no sequin annotation was provided to StringTie Mix or TASSEL at the merge stage for any of the comparison points here. **D.** Number of total matching sequin transcripts assembled by the indicated assembly method with ONT PCR-cDNA sequencing kit. **E.** Sensitivity of the indicated assembly methods at the locus level for the HAP1 dataset. Transcriptome assembled by the given method in the HAP1 dataset was compared against reference annotation (gencode hg38v35) using gffcompare. SLURP-stranded reads were used for FLAIR assembly. Apparent sensitivity indicates use of gencode annotation as ground truth. **F.** Percent of assembled transcripts that match completely with a transcript (left) or are contained within an intron (right) of the reference transcript, using only long-read assembly, StringTie Mix or TASSEL in the HAP1 and HL1 datasets. **G.** Comparison of inclusion or omission of SLURP stranding in TASSEL. Shown are percentage of transcripts mapped to the opposite strand (*top*) or completely matching (*bottom*) the reference transcripts in the HAP1 dataset. **H.** Number of transcripts contributed by short- and long-read assemblies to the TASSEL transcriptome in the sequin and HAP1 datasets. Long or short-read transcripts were used as a query against the TASSEL merged transcriptome as a reference in gffcompare. **I.** Proximity of TSS (left) and TTS (right) of known genes (gencode hg38v41) to the TSS and TTS of consensus transcripts in TASSEL or those of short or long only transcripts in the HAP1 dataset. **J.** Enrichment of H3K4me3 (left, normalized to input) and RNA Pol II (middle, normalized to the total number of mapped reads) and average PRO-seq signal (right) at the TSS of consensus transcripts in TASSEL or those of short or long only transcripts in the HAP1 dataset. H3K4me3 and RNA Pol II occupancy was calculated from ChIP-seq data from HAP1 cells [97,103]. Normalized PRO-seq data in HAP1 cells were obtained from [99]. **K.** Example of a gencode hg38v41 transcript assembled by TASSEL where the short- or long-read assembly failed to do so. **L.** Sensitivity and precision of sequin transcript assembly using short-read, long-read or TASSEL assembly with long read data from the PacBio sequencing platform. **M.** Relative depth of short to long read coverage

(mapped bases) in the indicated datasets.
(PDF)

**S6 Fig. Supporting data for Fig 6. A.** Average count of genes in the chromatin and nucleo-plasm fractions in HAP1 (left) and HL1 (right). Normalized gene counts obtained through DESeq2 were averaged for replicates (n = 3). **B.** Gene ontology enrichment analyses of protein-coding genes closest to cheRNA genes detected in HAP1 and HL1 datasets. Top 15 categories detected by DAVID [101] and associated number of genes are shown. Color bars correspond to false discovery rate (FDR). **C.** Histogram depicting the extent of overlap between cheRNA genes from short-read assembly and genes assembled from long-read assembly in HAP1 and HL1 samples. **D.** Comparison of the abundance (normalized DESeq2 gene count) of cheRNA genes showing minimum (0–10%) and maximum (90–100%) overlap with a corresponding long-read gene. Data points outside 1.5x of Inter-quartile range were removed as outliers. **** $p < 0.0001$, Two-tailed Mann-Whitney test. **E.** Prevalence of mono and multi-exonic cheRNA genes in segmented or unsegmented cheRNA genes. **F.** Metagene plot depicting average RNA Pol II occupancy (solid lines; shaded regions indicate SE) at the TSS ($\pm$1Kb) of segmented, unsegmented and TASSEL-refined cheRNA genes in the HL1 dataset. RNA Pol II occupancy was calculated from ChIP-seq data from heart of an eight-week-old mouse [102].
(PDF)

**S1 Note. Long-read alignments capture novel cell line-specific gene fusion events.**
(DOCX)

**S1 Table. Sequencing output of short- and long-read RNA-seq.**
(PDF)

**S2 Table. Primers used for RT-qPCR analyses.**
(PDF)

## Acknowledgments

We thank Peter Faber, Lindsay Scarpitta, and Mikayla Marchuk in the University of Chicago Functional Genomics Facility for Illumina sequencing. We thank Dr. Jonathan Göke and other members of the SG-Nex consortium for providing relevant files of sequin annotations. We thank members of the Ruthenburg and Ivan Moskowitz labs at the University of Chicago for helpful suggestions to improve the manuscript.

## Author Contributions

**Conceptualization:** Alexander J. Ruthenburg.

**Data curation:** Amoldeep S. Kainth, Gabriela A. Haddad, Johnathon M. Hall.

**Formal analysis:** Amoldeep S. Kainth, Gabriela A. Haddad.

**Funding acquisition:** Alexander J. Ruthenburg.

**Investigation:** Amoldeep S. Kainth, Gabriela A. Haddad.

**Methodology:** Amoldeep S. Kainth, Gabriela A. Haddad.

**Project administration:** Alexander J. Ruthenburg.

**Resources:** Alexander J. Ruthenburg.

**Software:** Amoldeep S. Kainth.

**Supervision:** Alexander J. Ruthenburg.

**Visualization:** Amoldeep S. Kainth, Gabriela A. Haddad, Alexander J. Ruthenburg.

**Writing – original draft:** Amoldeep S. Kainth, Gabriela A. Haddad, Alexander J. Ruthenburg.

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
