## [Decision Letter · Decision Letter 0]

18 May 2023

Dear Prof Ruthenburg,

Thank you very much for submitting your manuscript "Merging short and stranded long reads improves transcript assembly" for consideration at PLOS Computational Biology.

As with all papers reviewed by the journal, your manuscript was reviewed by members of the editorial board and by several independent reviewers. In light of the reviews (below this email), we would like to invite the resubmission of a significantly-revised version that takes into account the reviewers' comments.

The major concerns of the reviewers, which will need to be sufficiently addressed in a revision, center around the evaluation of the transcriptome assemblies. In particular, there were concerns that (1) the precision of TASSEL is not appropriately evaluated (2) the straightforward use of a reference annotation as a ground truth is misleading and (3) use of ERCC RNA standards only allows for the evaluation of assembly performance for monoexonic transcripts. I have a few comments in addition to those of the reviewers that I hope you would also address in a revision:

1. Given the emphasis of PLoS Computational Biology on computational methods, it would be helpful to have additional experiments that explore the individual contributions of the novel components of TASSEL to its performance. The two main components appear to be (A) the stranding of long reads with SLURP and (B) the strand-aware merge of the long and short-read assemblies. Computational experiments that leave out one or both of these components could be performed to determine how performance is affected. For example, how does TASSEL perform without the SLURP stranding? Or, how well does TASSEL perform with only the short-read data, or only the long-read data? If you provide SLURP stranded long-reads to the other assemblers, does that also improve their performance?

2. Related to the reviewers' concerns regarding the evaluation, I am concerned about the validity of using a reference annotation as a comparison for the output of "guided" assemblers, which use that same reference annotation as an input (i.e., the results in Figure S2A). Perhaps there are alternative evaluation methods that could be used.

3. Figure 2F - should the x-axis label be "% of short-read transcripts" or "% of long-read transcripts"? The text corresponding to this figure is contradictory (lines 261-265).

We cannot make any decision about publication until we have seen the revised manuscript and your response to the reviewers' comments. Your revised manuscript is also likely to be sent to reviewers for further evaluation.

Sincerely,

Colin Dewey

Guest Editor

PLOS Computational Biology

Ilya Ioshikhes

Section Editor

PLOS Computational Biology

Lucy Houghton

Staff

PLOS Computational Biology

The major concerns of the reviewers, which will need to be sufficiently addressed in a revision, center around the evaluation of the transcriptome assemblies. In particular, there were concerns that (1) the precision of TASSEL is not appropriately evaluated (2) the straightforward use of a reference annotation as a ground truth is misleading and (3) use of ERCC RNA standards only allows for the evaluation of assembly performance for monoexonic transcripts. I have a few comments in addition to those of the reviewers that I hope you would also address in a revision:

1. Given the emphasis of PLoS Computational Biology on computational methods, it would be helpful to have additional experiments that explore the individual contributions of the novel components of TASSEL to its performance. The two main components appear to be (A) the stranding of long reads with SLURP and (B) the strand-aware merge of the long and short-read assemblies. Computational experiments that leave out one or both of these components could be performed to determine how performance is affected. For example, how does TASSEL perform without the SLURP stranding? Or, how well does TASSEL perform with only the short-read data, or only the long-read data? If you provide SLURP stranded long-reads to the other assemblers, does that also improve their performance?

2. Related to the reviewers' concerns regarding the evaluation, I am concerned about the validity of using a reference annotation as a comparison for the output of "guided" assemblers, which use that same reference annotation as an input (i.e., the results in Figure S2A). Perhaps there are alternative evaluation methods that could be used.

3. Figure 2F - should the x-axis label be "% of short-read transcripts" or "% of long-read transcripts"? The text corresponding to this figure is contradictory (lines 261-265).

Reviewer's Responses to Questions

**Comments to the Authors:**

Reviewer #1: In this paper, Kainth and Haddad et al create TASSEL, a pipeline that combines short and long read RNA seq data for transcriptome assembly. They highlight the unique advantages of each method and by leveraging each of their strengths they claim they can improve the sensitivity and accuracy of transcript assembly. TASSEL is based on the preexisting transcript assembly tool Stringtie2 which was used for assembling both data types and merging the transcriptomes. In the process of creating this pipeline they identified and subsequently removed a prevalent direct cDNA artifact. They also develop SLURP, which attempts to strand direct cDNA reads by using the Nanopore sequencing adapters that are theoretically present at the end of each read. To benchmark their pipeline they tested it on cheRNA data that they produced and validated their transcript assemblies using external TSS and TES databases and by performing H3K4me3, RNA Pol II enrichment, and PRO-seq. This work includes an in depth comparison between short and long-read technologies, as well as comparisons to contemporary long-read transcript assembly tools. However I do believe there are a few aspects that are missing in the evaluation of TASSEL in order for it to be claimed to be an improvement over other tools. Many of the figures, and legends, and methods can be improved in order to communicate exactly what was done and allow for reproducibility. I have included my major and minor comments below which I believe will help make the claims in this manuscript more robust.

Major comments

1. Evaluation

Crucially, it is claimed that TASSEL uses the advantages of both types of RNAseq data, however it is also possible that it combines the disadvantages of both data too. One example is the challenge of performing transcript assembly using short-reads, which can lead to the construction or combination of exons that do not exist. This is especially problematic for long multi-exon transcripts and can result in reporting transcripts that are not present in the sample. The rate of false positives is however not measured in TASSEL and it is possible that by combining these two datasets, the precision of the output is worse than other tools.

There are additional ways TASSEL should be evaluated:

* The precision of TASSEL needs to be evaluated on a complex transcriptome and not only the spike-ins. ERCC spike-ins are single-exon by design and therefore not a challenge for short-reads to construct. The writers should measure the precision of their merged output, this can be done in a few ways. Using more complex multi-exon spike ins, simulated data so that the underlying ground truth is known or remove a sample of annotations to act as a test set and attempt to identify them.

* TASSEL should not only be compared to long-read assembly tools but also tools that only do assembly using the short-read data. This should highlight the expected difference between the long-read and short-read exclusive methods and hopefully will show that TASSEL is the union of positive differences.

* Many of the comparisons and evaluations of TASSEL are measured in percentage, however because of coverage differences it is likely the absolute number of transcripts detected is very different between the short-read and long-read data. It would be good to know how many transcripts in the final merged assembly are derived from the short-read, the long-read or both as without these numbers it is possible one data source is dominating the contribution to the final merged output. The subsequent TSS and TES analysis should then be separated between transcript models that were detected solely from each method and from both to show the contribution of each of these tools to detecting accurate transcript ends.

2. Direct cDNA is discontinued

Two of the major advancements in TASSEL is the identification of a reverse transcription artifact and stranding of direct cDNA data. With the announcement that ONT is discontinuing the direct cDNA sequencing library preparation, can the authors comment on if their approach is applicable in other forms, such as to PacBio or PCR cDNA ONT sequencing.

3. Treatment of reads that are still unstranded after processing

By stranding the direct cDNA reads, the authors claim to reduce the occurrence of mirrored transcripts being called on both strands. It is not clear however what is done with reads that are unable to be stranded due to lacking unambiguous barcodes. Including these reads as stranded can incur false confidence in downstream applications. The authors note that “Transcripts that are still assembled on the opposite strand could be a result of a residual pool of reads that were not stranded” however it is not clear why they would keep these reads in the analysis. They should repeat the analysis by removing the reads that were unable to be stranded to confirm this hypothesis, and see if this leads to a better transcript assembly.

Minor comments:

* In the abstract line 23 it is stated that long-read transcript assembly lacks strand of origin, but this is not entirely true because dRNA is stranded, and there are library preparations that allow stranded sequencing.

* Figure 1h is unclear and could be explained better in the main text and legend. Figure D shows that at the ends of the transcript there is only a one directional drop before/after the TSS/TES, however figure 1h suggests there is a drop at both ends. Is this just showing the coverage of the read ends? If so this should be made clear. Furthermore the legend says there are heatmaps but there are none present.

* From Figure 2f the authors claim there is a convergence between the short and long read assemblies because there is an agreement of ~20%. To me an agreement of 20% is very low and would signify the opposite of a convergence. It could be there is some missing context here that explains why 20% is a convergence, if so the authors should include this.

* In Figure 2g a line is drawn signifying when the short-read coverage is close to the long-read coverage. The authors should include how this equal coverage is defined as there are several possible interpretations such as the same number of reads, the number of mapped bases, the average coverage across all bases etc.

* Also in Figure 2g it is claimed that depth is important however when comparing 100% sampling versus 10% where the coverage is similar, there is not a lot of difference in the measured metrics excluding the containment of the long-read, and therefore coverage does not seem to be a reason why short and long read assemblies are different. The authors should clarify this claim or better explain why they believe depth is important based on these results.

* Figure 5c should be made clearer in the legend that each point represents only the ERCCs at that concentration level as it can be hard to interpret the figure without realizing that.

Reviewer #2: This manuscript by Kainth et al proposes a methodology to combine short (Illumina) and long (Nanopore) reads transcriptome sequencing to improve the identification of transcripts. The authors show that Nanopore sequencing has a high rate of false antisense transcripts due to the lack of strand information of the Nanopore reads. They develop an analysis strategy that is able to correct the strand mapping of these reads. Then they combine short-reads and stranded long-reads based transcripts inferred by StringTie to provide a consensus transcriptome prediction (TASSEL) . The rationale of this approach is that the higher sequencing depth of the short reads, in comparison to long reads, make it possible to identify more transcripts. They compare their approach to StringTie mix, FLAIR and Bambu and use as benchmarking dataset the Lexogen ERCC set. Finally, they apply their method to the identification of chromatin enriched RNAs (cheRNAS) in two cell lines HAP1 and mouse HL1.

This work has two components, the development of TASSEL as a tool for the reconstruction of transcripts using long and short reads, and the analysis of cheRNAs using this method.

The false identification of antisense transcripts due to the lack of strand information of Nanopore reads is a known limitation of the technology and the development of a strategy to mitigate this is relevant contribution. However, their approach is based on the identification of cDNA primers, which according to their result was only successful for about half of the reads. This should be highlighted in the discussion. Moreover, I am concern with several of the aspects in the strategy they took to benchmark their approach.

1. The benchmarking is done based on ERCC, which are synthetic, not spliced transcripts with no alternative isoforms and they conclude on performance based on this data. However, not spliced, single isoform genes are rare in the human genome, therefore the good performance of their approach on the ERCC data cannot be extrapolated to the broad of the human transcriptome where alternative splicing and the presence of similar isoforms in each gene, is abundant. The SIRVs spike-in sets of Lexogen, which contained multiple alternative isoforms of the same genes, are a better benchmarking dataset which should be used. Otherwise, the authors should clearly state that they approach has only be validated on mono-exon genes.

2. The authors use multiple data sources to validate TTS and TSS of their predicted transcript models and they show that TASSEL recover those better than other approaches and conclude that the transcripts are better assembled. Again, this conclusion is valid for monoexon genes but not for alternatively spliced transcripts, where multiple combinations of intron chains and TSS/TTS are possible.

3. Related to the two previous comments. It is unclear how their method performs on multi-exons genes with several alternative isoforms, and this should be specifically addressed.

4. The authors claim that a problem of long reads data for transcript assembly is the lower sequencing depth, and they show several analyses to proof this. Their long reads dataset has 1-3 million reads. However, much higher sequencing depths (up to 10-20 million reads) on one MinIon are now possible with Nanopore sequencing. Will TASSEL still provide an advantage in this scenario?

5. Figure 5. Unclear how the sensitivity with the entire transcriptome as the ground truth is not known. The methods section indicates that this analysis is taking as ground truth the genome annotation, which is not an accurate option, as genome annotation is incomplete even in extensively studied genome as human, and not all annotated genes or transcripts are present in each cell line or tissue Therefore, sensitivity and precision cannot be calculated in this way. Either these metrics are re-named to reflect that the ground truth is unknown, of a methodology is used that contains the ground truth, for example, simulated data, spliced SIRV or a subset of manually curated transcripts.

Regarding the analysis of the cheRNA fraction of the two cell lines, I have some concerns.

6. Only the Chromatin fraction was sequenced by long reads. Since the manuscript indicates a low agreement (20-25%) between short and long reads transcripts, there is a risk that the cheRNAs detection is highly biased by the transcript detection properties of the long reads technology. In fact, their analysis indicates that only 20-40% of the chromatin genes obtained with short reads are covered by the long reads. This is attributed to the lower sequencing depth of the long reads but posible other capture issues might occur. Actually, Figure 2C shows that the great majority of the long reads transcripts are intron retention, which is not the case for the short-reads data. More clarity if needed to understand how the sequencing technology used in each fraction is impacting the biological conclusions of the analysis.

7. When comparing chromatin and nuclear fractions, the transcriptomes have to be merged. It is not clear how incomplete transcripts or unprocessed transcripts are considered.

Other points of revision are:

a. Authors conclude from Figure 2B Good correlation between replicates in long reads sequencing. However, the plot shows that correlation at the médium/low expression range might not be. Using one correction value along the whole expression range is misleading as the highly expressed genes pull the correlation coefficient to high values. This analysis should be revisited.

b. I disagree that Figure 2E shows a high correlation between long and short reads data. Moreover, it is not clear how the pairing on long and short transcripts is done for this graph. Are there only exact match considered? How is redundancy handled, i.e. when one long reads transcript is a perfect match by multiple short reads transcripts or vice versa.

c. The authors conclude on the absence of length quantification biases for short and long reads sequencing from ERCCs, which is a reduced set of transcripts. Can the same conclusion be extracted for the whole transcriptome?

d. I do not think that agreeing in a high convergence between short and long reads assembly can be drawn when only 20-25% of the long-reads transcripts are matched by a short-reads transcript (line 261).

e. Also, I do not think that agreeing in a high convergence between short and long reads assembly can be drawn when only 20-25% of the long-reads transcripts are matched by a short-reads transcript (line 261).

f. The authors found a very interesting artifact in the cDNA library preparation which they address in their analysis. However, this is not the only artifact of cDNA libraries. Other are RT switching (which is similar but only skips a part of the transcript) and intrapriming, where the oligodT primes at A stretches different from the polyA, for example at introns. I wonder how their pipeline handles these types of cDNA library preparation errors.

g. When comparing their method to Bambu and FLAIR, I believe that only Long reads data is used for these comparing approaches. I am not sure about Bambu, but I believe FLAIR can integrate short reads data as well. I would be fairer to compare their method to FLAIR using both short and long reads data.

h. I found figures 1F and 1D for the long reads data, give contradictory results. Figure 1F suggests a general lack of full-lengthness for long reads sequencing, as more coverage is seen at the transcript ends. However, the data from the ERCCs do not show that. Is there any explanation for this result?

Reviewer #3: The manuscript "Merging short and stranded long reads improves transcript assembly" describes the development of tools to overcome the problem of strand assignment of long reads and to combine long and short reads to improve the resolution of transcript models, and then applies the tools to the annotation of cheRNA transcripts. The manuscript is very clearly written and the analysis and presentation of results similarly thorough and clear. There is a very comprehensive introduction to the use of long transcriptomic data in transcript annotation and useful investigation of the artefacts that can be identified. The final study of cheRNAs demonstrates the utility of the tools described earlier.

There is a lack of discrimination between spliced and unspliced/monoexonic lncRNAs. While the manuscript indicates that it is the annotation of spliced transcripts that is being addressed eg Figure 2C it seems that much of the benefit of the tools described relates primarily to monoexonic lncRNAs. For example, while SLURP demonstrably improves the strand assignment for long reads, this is less of a problem for spliced transcripts which can be stranded using splice site information from aligned reads. Similarly the cheRNA study, where SLURP and TASSEL improve on the transcript annotation over methods using unstranded long reads or short reads alone, appear to be focussed on the annotation of monoexonic lncRNAs.

It would be very useful if this issue could be discussed in the manuscript with perhaps some analysis provided in the supplementary data to assess the impact of SLURP and TASSEL on monoexonic vs multiexonic lncRNAs.

There is also some scope to add more explanation and analysis regarding the annotation of TSS and TTS. Was any testing done of the relative contribution of long vs short reads to the ability of TASSEL to corectly identify transcript termini? For example, analysis earlier in manuscript presents the coverage of transcripts by both datatypes, but no specific analysis is presented eg what happens to the H3K4me3 occupancy, Pol II occupancy and PRO-seq signal plots when the reads from each data type are removed? Does all the TSS signal come from long reads?

**Have the authors made all data and (if applicable) computational code underlying the findings in their manuscript fully available?**

Reviewer #1: Yes

Reviewer #2: Yes

Reviewer #3: Yes

PLOS authors have the option to publish the peer review history of their article (what does this mean?). If published, this will include your full peer review and any attached files.

Reviewer #1: No

Reviewer #2: No

Reviewer #3: No
---

## [Decision Letter · Decision Letter 1]

29 Aug 2023

Dear Prof Ruthenburg,

Thank you very much for submitting your manuscript "Merging short and stranded long reads improves transcript assembly" for consideration at PLOS Computational Biology.

As with all papers reviewed by the journal, your manuscript was reviewed by members of the editorial board and by several independent reviewers. In light of the reviews (below this email), we would like to invite the resubmission of a significantly-revised version that takes into account the reviewers' comments.

I appreciate your thorough response to the prior reviewers' comments and, in particular, the addition of benchmarks with the sequin spike-in RNA standards. However, due to challenges associated with reviewer availability, an additional reviewer (Reviewer 4) was recruited to review the revised manuscript, and Reviewer 4 has raised some significant new concerns. In particular, Reviewer 4 has concerns about the novelty, need for, and evaluation of the computational pipeline in light of the existence of the ONT pychopper tool and other more recent methods that can clean and orient ONT reads. It is unfortunate that these major concerns are being raised at this stage of the review process, but they must be addressed before a decision is made regarding the suitability of this work for publication in PLoS Computational Biology.

We cannot make any decision about publication until we have seen the revised manuscript and your response to the reviewers' comments. Your revised manuscript is also likely to be sent to reviewers for further evaluation.

Sincerely,

Colin Dewey

Guest Editor

PLOS Computational Biology

Lucy Houghton

%CORR_ED_EDITOR_ROLE%

PLOS Computational Biology

I appreciate your thorough response to the prior reviewers' comments and, in particular, the addition of benchmarks with the sequin spike-in RNA standards. However, due to challenges associated with reviewer availability, an additional reviewer (Reviewer 4) was recruited to review the revised manuscript, and Reviewer 4 has raised some significant new concerns. In particular, Reviewer 4 has concerns about the novelty, need for, and evaluation of the computational pipeline in light of the existence of the ONT pychopper tool and other more recent methods that can clean and orient ONT reads. It is unfortunate that these major concerns are being raised at this stage of the review process, but they must be addressed before a decision is made regarding the suitability of this work for publication in PLoS Computational Biology.

Reviewer's Responses to Questions

**Comments to the Authors:**

Reviewer #1: In this response the authors have addressed all major and minor issues I raised after reading the initial submission. The addition of the analysis using sequins was a great addition to the manuscript and strengthens many of the claims the authors make with TASSEL. I also appreciate the significant developments TASSEL made, enabling its use on PCR cDNA ONT samples.

Reviewer #4: The authors propose a pipeline TASSEL to assemble transcripts from short and long dRNA and cDNA sequencing reads. The pipeline contains standard steps such as orienting the reads (with their tool SLURP), aligning the oriented reads with minimap2, and assembling transcripts with StringTie2. The authors forget to perform standard primer trimming with pychopper (standard to ONT long-read processing), which is probably also why FLAIR performs badly, as it expects preprocessing. The main findings presented, such as "long-read transcript assembly lacks depth and strand-of-origin" and "We also discover a cDNA synthesis artifact in long-read datasets" are therefore not true (pychopper can be used to fix both of these claims). Consequently, SLURP may be redundant compared to the previously designed tools for the same purpose.

On the positive side, the paper contains some informative analyses of lncRNA and cheRNA. Overall, the paper reads more like a biological analysis and would (after addressing below major comments) fit better in such a venue than a Computational Biology venue.

Major

1. L76-96. The authors discuss state-of-the-art tools but misses some of the work that is related most closely related to this paper. For the error-correction discussion, they miss TALC [1] (a hybrid tool). Other non-hybrid long-read approaches exist for ONT transcript assembly [2-7]. Long cDNA reorientation and chimeric removing tools exist: lima (PacBio), pychopper (ONT), and academic tools: Reorientexpress [8], mandalorian pipeline [9].

2. L313-316 and the following section: It seems the authors did not use pychopper (https://github.com/epi2me-labs/pychopper) to identify, orient and trim full-length reads before alignment, as is standard (or any of the already existing academic alternatives such as Reorientexpress [8], mandalorian pipeline [9]). Consequently, at best, SLURP perhaps improves over pychopper or other computational read-orienters but is not benchmarked. At worst, SLURP may be redundant.

3. Section starting at L377: "these analyses reveal a widespread heretofore unknown artifact" Identifying chimeric transcripts due to e.g. RT-PCR artifacts is standard in long-read primer trimmers such as pychopper for ONT and lima (https://lima.how/) for PacBio and such artefacts are well known. The authors should look up "full-length non-chimeric reads" (see eg https://www.ncbi.nlm.nih.gov/pmc/articles/PMC6590394/) which was a standard concept at least a few years ago. Most analysis tools are designed to process full-length non-chimeric reads after pychopper/lima has been applied.

4. As a consequence of points 2 and 3, sections starting at L577 and L601 are not novel findings. Pychopper should be used, which should fix both these issues.

Minor

- L126: clarify which types "other long-read datasets" refers to.

- L156: Missing reference to tool used for short read alignments.

- L321-L322: "inaccurate alignment of reads to the incorrect strand" - it is not the alignment that is inaccurate.

- L562: "we systematically compared multiple transcript assembly approaches". This is an overstatement, as many state-of-the-art methods are missing.

References

[1] Broseus L, Thomas A, Oldfield AJ, Severac D, Dubois E, Ritchie W. TALC: Transcript-level Aware Long-read Correction. Bioinformatics. 2020 Dec 22;36(20):5000-5006. doi: 10.1093/bioinformatics/btaa634. PMID: 32910174.

[2] Kuo, R.I., Cheng, Y., Zhang, R. et al. Illuminating the dark side of the human transcriptome with long read transcript sequencing. BMC Genomics 21, 751 (2020). https://doi.org/10.1186/s12864-020-07123-7

[3] Wyman, D. et al. A technology-agnostic long-read analysis pipeline for transcriptome discovery and quantification. Preprint at bioRxiv https://doi.org/10.1101/672931 (2020).

[4] Prjibelski, A. D. et al. Accurate isoform discovery with IsoQuant using long reads. Nat. Biotechnol. https://doi.org/10.1038/s41587-022-01565-y (2023).

[5] de la Rubia, I. et al. RATTLE: reference-free reconstruction and quantification of transcriptomes from Nanopore sequencing. Genome Biol. 23, 1–21 (2022).

[6] Nip, K.M., Hafezqorani, S., Gagalova, K.K. et al. Reference-free assembly of long-read transcriptome sequencing data with RNA-Bloom2. Nat Commun 14, 2940 (2023). https://doi.org/10.1038/s41467-023-38553-y

[7] Alexander J Petri, Kristoffer Sahlin, isONform: reference-free transcriptome reconstruction from Oxford Nanopore data, Bioinformatics, Volume 39, Issue Supplement_1, June 2023, Pages i222–i231

[8] Ruiz-Reche A, Srivastava A, Indi JA, de la Rubia I, Eyras E. ReorientExpress: reference-free orientation of nanopore cDNA reads with deep learning. Genome Biol. 2019 Nov 29;20(1):260. doi: 10.1186/s13059-019-1884-z. PMID: 31783882; PMCID: PMC6883653.

[9] Byrne A, Beaudin AE, Olsen HE, Jain M, Cole C, Palmer T, DuBois RM, Forsberg EC, Akeson M, Vollmers C. Nanopore long-read RNAseq reveals widespread transcriptional variation among the surface receptors of individual B cells. Nat Commun. 2017 Jul 19;8:16027. doi: 10.1038/ncomms16027. PMID: 28722025; PMCID: PMC5524981.

**Have the authors made all data and (if applicable) computational code underlying the findings in their manuscript fully available?**

Reviewer #1: Yes

Reviewer #4: Yes

PLOS authors have the option to publish the peer review history of their article (what does this mean?). If published, this will include your full peer review and any attached files.

Reviewer #1: No

Reviewer #4: No
---

## [Editor Report · Decision Letter 2]

5 Oct 2023

Dear Prof Ruthenburg,

We are pleased to inform you that your manuscript 'Merging short and stranded long reads improves transcript assembly' has been provisionally accepted for publication in PLOS Computational Biology.

Best regards,

Colin Dewey

Guest Editor

PLOS Computational Biology

Lucy Houghton

%CORR_ED_EDITOR_ROLE%

PLOS Computational Biology

---

## [Editor Report · Acceptance letter]

20 Oct 2023

PCOMPBIOL-D-22-01812R2 

Merging short and stranded long reads improves transcript assembly

Dear Dr Ruthenburg,

I am pleased to inform you that your manuscript has been formally accepted for publication in PLOS Computational Biology. Your manuscript is now with our production department and you will be notified of the publication date in due course.

With kind regards,

Anita Estes
